# Archaeal and eukaryotic MCM rings sequentially melt DNA for replication initiation

Sanaz Rasouli [1,4,5], Alexander Myasnikov [1,5] & Eric J. Enemark [1,2,3] ✉

DNA replication initiation requires local melting of fully base-paired DNA for a helicase to gain a foothold and initiate processive DNA unwinding. In eukaryotes and archaea, the helicase engine is the hexameric ring minichromosome maintenance (MCM) complex. In eukaryotes, a defined biochemical sequence assembles two Cdc45-MCM-GINS (CMG) complexes that provide limited DNA unwinding as the species that immediately precedes extensive unwinding. A prior structure revealed how MCM subunits interact with this form of DNA, but the atomic progression from undistorted DNA to this melted DNA species is unknown. Here, we present a sequential DNA melting mechanism determined by snapshots of an archaeal MCM ring with DNA in varying degrees of melting. In this mechanism, successive ATP-binding at MCM ATPase sites drives sequential discrete DNA melting steps mediated a specific MCM aromatic residue. Analysis of eukaryotic structures shows loaded MCM rings principally adopt only two molecular arrangements at the ATPase: one that does not melt DNA and one tuned to melt DNA with equivalent aromatic residues, indicating a universal sequential mechanism melts DNA in archaea and eukaryotes for replication initiation.

DNA replication is a fundamental, intrinsic life process of duplicating genetic material to enable cell division. The act of DNA replication proceeds with a forked DNA structure where a helicase enzyme separates the constituent strands of the DNA double-helix, and polymerases use the exposed single strands as templates in the synthesis of new DNA[1]. The DNA replication process is tightly regulated by the cell cycle to preserve genomic integrity by ensuring that DNA is fully replicated once, and only once, each cell cycle[2]. An essential step towards regulated replication initiation is the establishment of a hexameric ring helicase to encircle one DNA strand while excluding the other[3]. Once established in this topology, the ring helicase propagates strand separation as it moves unidirectionally along the DNA. Establishment of the active helicase species, where the helicase ring encircles only one DNA strand, requires that the two DNA strands locally

separate, termed melting, such that some base-pair interactions are lost. Bacteria first melt DNA with the initiator protein complex DnaA[4–7], which allows subsequent loading of the hexameric DnaB ring helicase[8,9] to encircle one of the exposed DNA strands[6,7,10,11]. In contrast, the sequence is reversed in eukaryotes and archaea with the minichromosome maintenance (MCM) complex loaded to fully double-stranded DNA[12,13] followed by local DNA unwinding and helicase activation[14] to ultimately yield a strand-excluded species that unwinds DNA and is competent for its replication.

In eukaryotes, the MCM complex is a heterohexamer of six similar subunits, Mcm2–7[15]. All eukaryotic MCM proteins are highly similar in structure and sequence, especially in the ATPase domain[16]. The archaeal MCM complex is a homohexamer of six identical subunits[17] that is highly similar in structure to eukaryotic Mcm2–7 with very

[1]Department of Structural Biology, St Jude Children's Research Hospital, Memphis, TN, USA. [2]Department of Biochemistry and Molecular Biology, University of Arkansas for Medical Sciences, Little Rock, AR, USA. [3]Winthrop P. Rockefeller Cancer Institute, University of Arkansas for Medical Sciences, Little Rock, AR, USA. [4]Present address: Phenomenex Inc., Torrance, CA, USA. [5]These authors contributed equally: Sanaz Rasouli, Alexander Myasnikov. ✉e-mail: ejenemark@uams.edu

strong sequence conservation in a 152-amino acid ATPase core that contains nearly all residues involved in both DNA-binding and ATP-binding[16]. MCM proteins belong to a large AAA+ family of ATPases[18]. These ATPases often function as motors that use energy derived from ATP hydrolysis to drive mechanical motion. The ATPase active sites are formed at subunit interfaces, analogous to the active sites of F1-ATPase[19]. One subunit contributes Walker-A and Walker-B residues, and the other contributes residues such as an arginine finger[20,21]. Based on analogy to F1-ATPase[19,22], papillomavirus E1 helicase[23], and SV40 large T-antigen[24], tight subunit interfaces are expected to correlate with ATP-binding, while loose interfaces are expected to correlate with the absence of nucleotide at the ATPase site. AAA+ helicases include MCM and the hexameric ring helicases of some viruses, such as papillomavirus E1[23,25], SV40 large T-antigen[24,26,27], and AAV Rep[28]. These helicases all form hexameric rings that project structural modules into the central channel to interact with DNA[23,24,26,29–31]. All of these AAA+ helicases use a pre-sensor-1 beta-hairpin (ps1β), named because its amino acids directly precede the AAA+ sensor-1 sequence motif[32]. The MCM helicases use an additional DNA-binding hairpin known as the helix-2-insert (h2i). This hairpin is a specialized insert in helix α2 of the AAA+ topology[18] that defines one AAA+ family clade[33]. We have previously shown that the h2i can adopt either a helix or horseshoe conformation in the α2 region that is N-terminal to the insert hairpin[31]. The helix conformation of an h2i develops a parallel β-sheet interaction with the ps1β of an adjacent subunit that is not present in the horseshoe conformation[31]. As a result, subunits with the helix conformation have a tighter subunit interface than those in the horseshoe conformation.

The defined sequence for replication initiation in eukaryotes begins with recognition of a DNA replication origin by the origin recognition complex (ORC)[2,14,34] that recruits the MCM complex. Initial recruitment of MCM rings by ORC yields a salt-sensitive species where the association of MCM at the origin relies on its association with ORC[35]. This species is converted to a salt-stable species, termed loaded, where the MCM:DNA association is independent of ORC[35,36]. This provides an Mcm2–7 double-hexamer[30] that encircles duplex DNA[37] and can passively slide on DNA without unwinding it[12,13]. Loading of the Mcm2–7 ring involves opening the Mcm2/5 interface known as the gate[21]. At the onset of S-phase, the Mcm2–7:DNA species is activated in a sequential series[38] where it is converted from a stable double-hexamer that encircles both DNA strands to single hexamers that each encircle only one strand[3]. During this activation, Cdc45 is recruited to each MCM ring to seal the Mcm2/5 gate[39]. Cdc45 recruitment does not generate detectable DNA unwinding[38]. Next, the tetrameric GINS complex is recruited adjacent to Cdc45 to generate a larger CMG (Cdc45-MCM-GINS) complex[40,41]. GINS recruitment provides 0.6–0.7 turns of DNA unwinding at each CMG[38] as the final detectable species that precedes the processive DNA unwinding of a replication fork.

Archaeal replication initiation is functionally similar to eukaryotic initiation. Archaeal ORC is homologous to eukaryotic ORC and binds at a DNA replication origin[42,43], and two MCM hexamers assemble as a double hexamer[17,44,45] that is highly similar in architecture to the eukaryotic Mcm2–7 double hexamer[30]. Archaea also possess GINS and Cdc45 homologs[46,47] that stimulate the activities of MCM in biochemical DNA-unwinding assays[48–50]. Crystal structures of the archaeal proteins[51–55] illustrate that they are indeed structural homologs of eukaryotic Cdc45[56] and GINS[57–59], suggesting fundamentally conserved mechanisms of action. For both archaea and eukaryotes, the MCM hexamer ring by itself is sufficient to unwind DNA in a biochemical strand displacement assay[17,60], indicating general DNA unwinding is not the essential role of Cdc45 or GINS. Instead, eukaryotic Cdc45 and GINS are essential for the initial melting of DNA at a replication origin for initiation[38] and are essential to establish origin-dependent DNA replication[61]. The precise roles of archaeal Cdc45 and GINS are not known, but presumably they function analogously to the eukaryotic

proteins to enable specific activation of the MCM helicase at a replication origin.

Functionally, MCM has very strong similarities with the viral SuperFamily 3 (SF3) helicases[62,63] papillomavirus E1 and SV40 large T-antigen (L-tag). Each assembles at a replication origin as a double-hexamer and unwinds duplex DNA bidirectionally once activated[64,65]. Early electron microscopy images showed that MCMs and SF3 helicases form highly similar dumbbell-shaped double-hexamers[17,66,67]. The underlying double-hexamer architecture is highly similar to that of MCM with the smaller N-terminal domains forming a head-to-head interaction at the middle and the larger ATPase domains located at each side[17,67]. All exhibit a 3′->5′ polarity when unwinding DNA[17,68,69]. The ATPase domains share strong sequence homology and belong to the AAA+ family of ATPases[18]. All include a ps1β that interacts with the sugar-phosphate of the DNA backbone[23,27,31]. In the case of E1, the ps1β has two distinct functional residues: (1) a conserved lysine is essential for processive DNA-unwinding; and (2) an aromatic residue that is dispensable for DNA-unwinding is essential for DNA melting and DNA replication[70].

The sequential generation of CMG requires prior exhaustion of ADP from the MCM ATPase sites followed by ATP-binding, but not ATP-hydrolysis[38]. Similarly, initial untwisting of DNA at each CMG relies upon ATP-binding, but not ATP-hydrolysis[38]. The cryo-EM structure of the species that immediately precedes processive unwinding reveals melted DNA and its interactions with the MCM subunits of CMG[71,72]. In particular, the ps1β and h2i of multiple adjacent ATPase domains interact with the melted DNA, principally with the strand that will subsequently be encircled to become the leading strand DNA template[71]. In this structure, the ATPase modules that interact with DNA belong to Mcm3, Mcm5, Mcm2, and Mcm6[71], which are adjacent subunits in the ring at the same side of the ring where Cdc45 and GINS are recruited[71]. The atomic progression from undistorted DNA to this melted DNA species is unknown because the MCM:DNA structures that precede this step, including the nucleotide-free species of the first step, are not known.

Here, we show that DNA melting is mediated by MCM ATPase domains in a stepwise mechanism. We provide multiple high-resolution snapshots of DNA melting obtained by single particle cryo-EM analysis of an archaeal MCM encircling a segment of duplex DNA. The melting stages are driven by an aromatic residue on the h2i of three consecutive MCM subunits that collectively form an aromatic wedge. The subunits that participate in melting require a helix conformation in the h2i, which in turn correlates with a tight subunit interface and an ATP mode of binding. The series of structures collectively suggest progressive DNA-melting by an MCM aromatic wedge driven by successive ATP-binding. The final structure in the series strongly resembles the structure of eukaryotic CMG with melted DNA[71], suggesting a universal structure for the species that immediately precedes processive DNA unwinding by MCM. The arrangement of the aromatic residue with DNA is highly similar to that of E1 and L-tag, suggesting that an aromatic wedge mechanism to melt DNA for replication initiation is universal among archaea, eukaryotes, and DNA tumor viruses.

## Results and discussion

With single particle cryo-EM (Supplementary Figs. 1–9 and Supplementary Tables 1–6), we characterize multiple distinct MCM:DNA structures (Fig. 1) that are consistent with a sequential DNA melting mechanism mediated by a conserved aromatic residue of the h2i. The DNA used for the structure determinations are fully double-stranded, indicating that lost base-pairs correspond to bona fide DNA melting. The protein construct is a chimeric fusion of the *Saccharolobus solfataricus* MCM N-terminal domain and the *Pyrococcus furiosus* MCM AAA+ domain, *Sso-Pf*MCM. We have previously determined the hexameric ring crystal structure of this construct and shown that it has a robust strand displacement activity[29]. This *Sso-Pf*MCM chimera and a modified *Sso*MCM construct are the only two constructs that we have successfully crystallized in a hexameric form containing both the

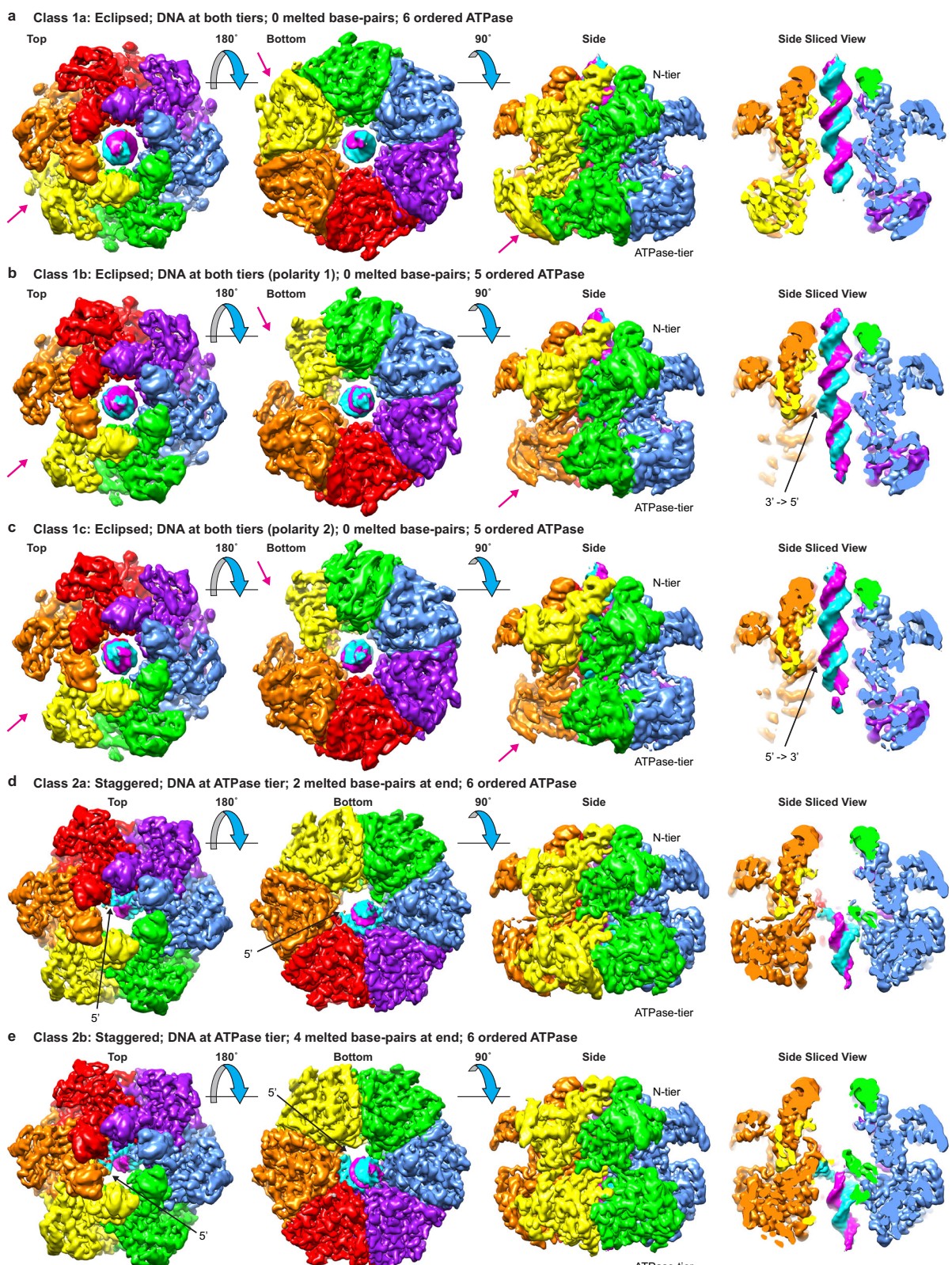

**Fig. 1 | Overall architectures of the MCM hexamer:DNA complexes.** Major class 1 eclipses the two tiers and encircles duplex DNA at both tiers. Major class 2 staggers the two tiers and encircles partially melted DNA at the ATPase tier. **a** Class1a: all 6 ATPase domains are well-ordered, including the yellow subunit (magenta arrow). **b** Class 1b: the ATPase domain of the yellow subunit (magenta arrow) is poorly ordered. The 3′->5′ DNA strand (cyan) is near the orange h2i (right panel). **c** Class 1c: the ATPase domain of the yellow subunit (magenta arrow) is poorly ordered. The 5′->3′ DNA strand (magenta arrow) is near the orange h2i (right panel). **d** Class2a: two terminal base-pairs are melted with the 5′-end of the cyan DNA strand near the red ATPase domain hairpins. **e** Class2b: four terminal base-pairs are melted with the 5′-end of the cyan DNA strand near the orange ATPase domain hairpins. Unsharpened maps are illustrated to emphasize the general MCM:DNA features.

N-terminal and AAA+ domains. We previously reported cryo-EM structures of the modified *Sso*MCM construct encircling single-stranded DNA, which allowed us to identify additional states of DNA translocation[16]. For this study, we were motivated to elucidate how the archaeal MCM hexamer interacts with encircled double-stranded DNA, something we have never been able to capture by X-ray crystallography. The *Pyrococcus furiosus* genetic sequence has an intein in the AAA+ domain (*Pf*MCM amino acids 362–728[48]) that was genetically removed from the construct[29]. In this report, residues are numbered with native amino acid numbers without intein removal, such than residue 361 is directly linked to residue 729.

To verify that the MCM:DNA structures are general and not associated with a specific DNA sequence or DNA reagent, structures were determined for samples generated from two independent DNA sequences. The underlying structural classes of these two samples are highly similar, which enabled particle merging to generate higher resolution versions of each structural class. For each MCM:DNA species, the multiple structures were determined from a single cryo-EM grid as different 3-dimensional classes. These consist of single-hexamers encircling DNA and illustrate stepwise melting of DNA at the MCM ATPase tier and not the potential attributes of two hexamers together. All the structural classes adopt expected two-tiered ring-shaped architectures described previously[12,17,30,37,41,73–76] with DNA located inside the central channel of the ring. One tier consists of the N-terminal domain (N-tier) in three subdomains defined previously[44], consistent with many other MCM structures[44,77,78]. The other tier (C-tier) consists of the AAA+ family ATPase domain[18,79,80]. Nearly all structural classes adopt an expected hexameric architecture. A minor fraction of heptameric MCM rings encircling DNA are observed. Heptameric MCM rings have been observed previously[45,74,81,82], but the heptameric MCM form is not expected to function biologically and will not be a focus of subsequent discussion.

Our major focus is the arrangement of the ATPase tier and its DNA-binding hairpins with DNA. Relative to this portion, the N-tier occupies two distinct rotational orientations when viewed down the central channel that define two Fundamental Classes. The distinct orientations will be referred to as eclipsed (Fundamental Class 1) and staggered (Fundamental Class 2) based on the alignment of one tier subunit:subunit interface with the corresponding subunit:subunit interface of the other tier. Inter-tier orientation differences have been described previously in comparing ORC/Cdc6/Cdt1/MCM2–7 to either Mcm2–7 double-hexamer or CMG[83], and we will expand this comparison in a section below. The two forms exhibit three major differences (Fig. 1 and Supplementary Figs. 10 and 11): (1) the central channel diameter of the ATPase tier is significantly wider in eclipsed Fundamental Class 1 versus staggered Fundamental Class 2; (2) duplex DNA is observed within the central channel of both tiers in eclipsed Fundamental Class 1 but is only observed at the ATPase tier of staggered Fundamental Class 2; and (3) the DNA is fully base-paired in the eclipsed hexamer, but a portion of the DNA in the staggered hexamer is single-stranded, indicating that DNA that was fully duplex during sample preparation had become melted.

## Tight and loose subunit interfaces

Each fundamental class encompasses multiple subclasses. Collectively, the subclasses provide a framework for sequential melting of DNA based on the ATPase tier. All the MCM ATPase sites of the structures indicate the presence of bound nucleotide with varying strengths. The Walker-A motif of an ATP-binding site binds anions, particularly sulfate or phosphate, or nucleotides if these are in sufficient concentration. Thus, the presence of nucleotide at these sites is anticipated at the high 5 mM Mg/ATP-γS concentration used during sample preparation. We use two intersubunit criteria that are readily discerned in the cryo-EM maps to assign whether the interface is tight or loose and define the nucleotide status of each ATPase site. Interfaces that satisfy both

criteria are assigned as tight. This differentiation method is conceptually similar to those applied in other helicase structural studies, such as bacteriophage T7 that differentiated ATPase site types by crystallographic occupancy[84] and papillomavirus E1 that differentiated site types by the chemical architecture of the sites[23]. Both prior studies did not discern a formally different molecule bound in the site.

The first criteria we use to assign a tight subunit interface is the relative proximity of modules that comprise the bipartite ATPase site. Tight interfaces have a consistent and small appearance for a vacant volume (labeled T in Fig. 2) near a clearly visible γ-S-phosphate. The corresponding vacant volume of loose interfaces (labeled L in Fig. 2) is larger and variable in appearance. We adopt an intersubunit distance threshold of 6.8 Å for a P-loop proline Cα of one subunit to a Cα of the arginine finger motif (four residues prior to the conserved arginine) of the neighboring subunit (Fig. 2 and Table 1) for this tight interface criteria.

The second feature we use to define a tight subunit interface is the proximity of the h2i hairpin of one subunit to the ps1β of the neighboring subunit (Fig. 3). We have previously illustrated that the portion of helix-2 that faces the central channel (residues N-terminal to the insert) can adopt a traditional helix conformation or a horseshoe conformation[31]. The helix conformation binds the DNA sugar-phosphate backbone with a fully conserved hydroxyl side chain, but the horseshoe conformation places this hydroxyl group too far from the DNA for direct interaction[31]. The horseshoe conformation also necessitates a loose subunit interface because the horseshoe structure would clash with the neighboring ps1β within the tight interfaces that are developed when adjacent DNA-binding hairpins interact with consecutive DNA nucleotides (Fig. 3). Analogous to the ATPase site distance, we have adopted a distance threshold of 8.0 Å for a specific pair of h2i-ps1β residues (a conserved h2i leucine and a conserved ps1β lysine, Fig. 3 and Table 1) for this tight interface criteria.

Based on these criteria (Fig. 3 and Table 1), all subunit interfaces of the Class 1 structures are loose. The Class 2a structure has 2 tight interfaces and 4 loose interfaces, and Class 2b has 3 tight interfaces and 3 loose interfaces (Figs. 2 and 3 and Table 1).

## Comparative analysis I: general inter-tier dihedrals quantify eclipsed versus staggered

The different inter-tier conformations of the eclipsed and staggered forms is qualitatively similar to the inter-domain orientation differences described previously in comparing ORC/Cdc6/Cdt1/MCM2–7 to either the Mcm2–7 double-hexamer or CMG[83]. In particular, prior analysis described in detail multiple MCM:Cdt1 interactions in driving different inter-domain orientations[83]. To quantitatively compare the inter-tier conformations of our structures to eukaryotic complexes, we calculated a dihedral angle at each subunit of a complex[85,86] along with the mean dihedral angle. The NTD subunit position selected for the dihedral (large sphere in Fig. 4a inset) is near a subunit interface on a conserved loop[87,88] at the bottom of the NTD. The CTD subunit position selected for the dihedral (second large sphere in Fig. 4a inset) is near the same subunit interface at the top of the CTD. With these definitions, a 0° dihedral (perfectly eclipsed) approximately generates a continuous vertical interface seam across the N-tier and C-tier. Class 1a has a mean dihedral of +3 that approximately eclipses the interface seams of the two tiers. In contrast, Class2a and Class2b have mean dihedral angles of −32° and −31° that dramatically offset two interface seams in a staggered conformation. The different seam alignments for the Fundamental Classes can be viewed at the sides of the green subunits in the side views of Fig. 1, approximately along the horizontal midline of the molecule.

This algorithm was extended to all eukaryotic MCM complex structures of the PDB that contain six AAA+ domains and six OB-folds (72 total eligible)[85]. This analysis (Fig. 4) illustrates that all eukaryotic MCM structures fall into two obviously different categories (category 1 mean dihedral range: −6 to −15; category 2 mean dihedral range: −29 to

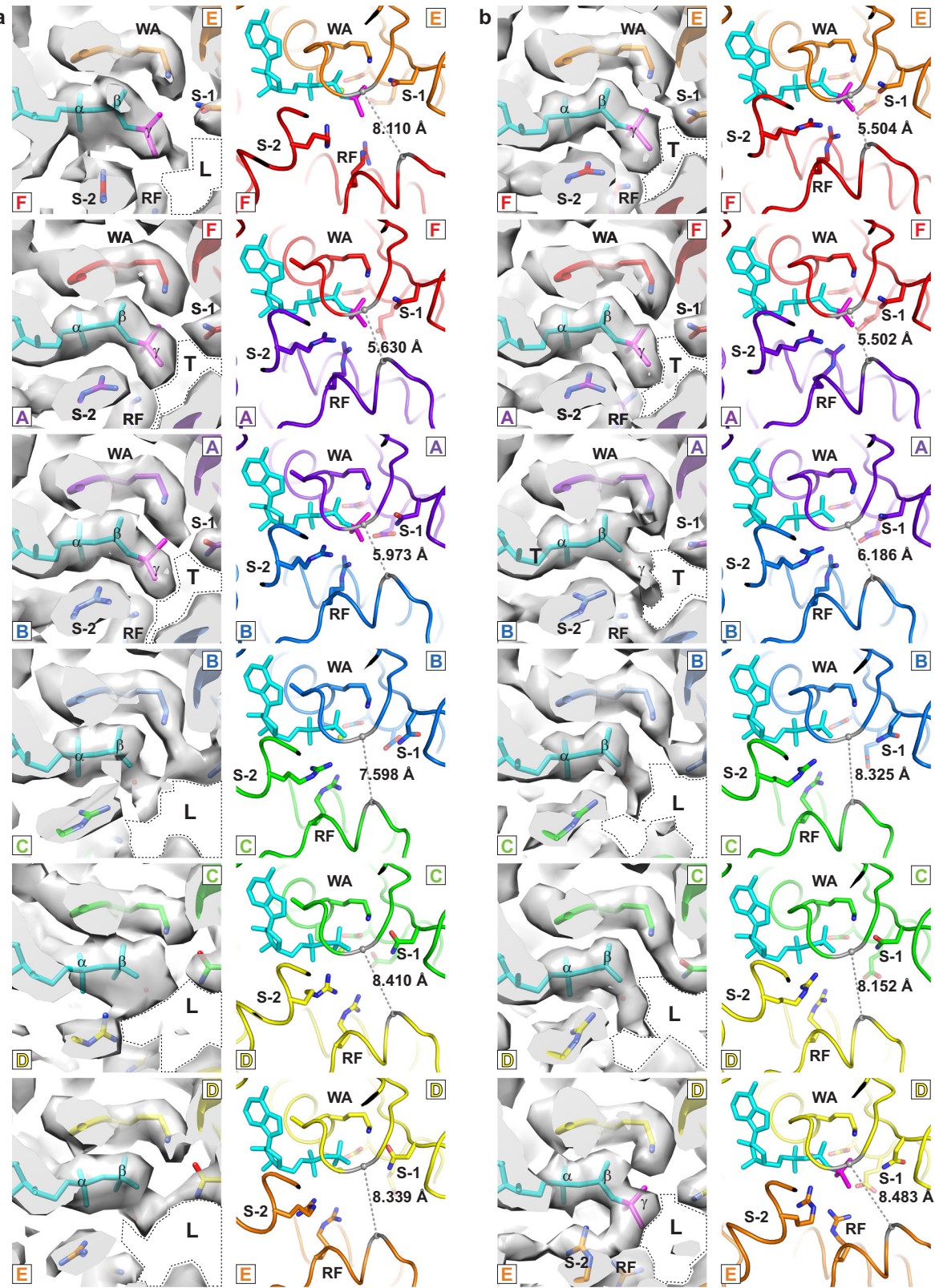

**Fig. 2 | Tight versus loose ATPase site architectures.** Each panel provides a consistent view of the bipartite ATPase site (WA: Walker-A; S-1: Sensor-1; R-F: arginine finger; S-2: Sensor-2). For each structure, the left panel shows the density with 50% transparency, including the strength of bound nucleotide γS-phosphate. A void space near sensor-1 and the arginine finger that contrasts tight (T) and loose (L) interfaces is emphasized with a dashed outline. The right panel of each structure depicts the molecular model. P330 Cα and T824 Cα positions are illustrated in grey spheres. A P330 Cα - T824 Cα distance <6.8 Å is one of the criteria used to assign a Tight interface (Table 1). **a** Class 2a shows two tight interfaces with a distinct, consistent shape for the highlighted void volume and a short intersubunit distance. **b** Class 2b shows three tight interfaces with a void volume and distance that are consistent with the tight interfaces of Class 2a. Chain identifiers of the coordinate file are provided for each panel (chain A: purple, chain B: blue; chain C: green; chain D: yellow; chain E: orange; chain F: red) in a color scheme consistent with Fig. 1.

**Table 1 | Intersubunit distances to define a tight interface**

| Structure: | Subunit interface: | | | | | | | | | | | | | DNA melted at ATPase |
|---|---|---|---|---|---|---|---|---|---|---|---|---|---|---|
| | D:E or 7:3 | | C:D or 4:7 | | B:C or 6:4 | | A:B or 2:6 | | F:A or 5:2 | | E:F or 3:5 | | |
| | ATP site | h2i-ps1β | ATP site | h2i-ps1β | ATP site | h2i-ps1β | ATP site | h2i-ps1β | ATP site | h2i-ps1β | ATP site | h2i-ps1β | |
| Class 1a | 8.713 | 16.351 | 8.959 | 16.46 | 9.043 | 16.19 | 8.961 | 16.471 | 8.828 | 16.305 | 8.913 | 16.487 | No |
| PDB 7W1Y[99] | 5.219 | 7.496 | 5.059 | 7.341 | 5.597 | 7.279 | 9.454 | 19.985 | 8.387 | 12.378 | 5.729 | 7.134 | No |
| PDB 5BK4[37] | 7.049 | 8.594 | 6.337 | 8.68 | 6.695 | 7.691 | 9.071 | 21.671 | 10.281 | 12.446 | 7.321 | 10.877 | No |
| PDB 9GJW[89] | 7.414 | 7.166 | 5.497 | 8.547 | 5.176 | 7.684 | 5.696 | 7.147 | 7.368 | 15.688 | 10.504 | 16.735 | No |
| PDB 9E2X[98] | 9.086 | 13.195 | 9.344 | 7.121 | 7.015 | 6.746 | 5.259 | 7.74 | 6.712 | 14.124 | 9.03 | 14.331 | N/A |
| Class 2a | 8.339 | 13.956 | 8.41 | 13.9 | 7.598 | 12.423 | 5.973 | 7.491 | 5.63 | 7.662 | 8.11 | 14.953 | 2 base-pairs |
| PDB 7PMN[96] | 7.744 | 11.603 | 6.125 | 12.507 | 7.244 | 7.126 | 5.217 | 7.268 | 5.219 | 7.235 | 8.511 | 21.045 | 2 base-pairs |
| Class 2b | 8.483 | 15.448 | 8.152 | 14.074 | 8.325 | 12.656 | 6.186 | 7.362 | 5.502 | 7.479 | 5.504 | 7.859 | 4 base-pairs |
| PDB 7PMK[96] | 8.365 | 16.895 | 6.546 | 12.843 | 6.987 | 11.214 | 5.143 | 7.179 | 5.121 | 6.967 | 5.693 | 7.138 | 4 base-pairs |
| PDB 7Z13[71] | 8.488 | 14.821 | 7.964 | 12.973 | 9.244 | 13.964 | 6.733 | 7.426 | 5.861 | 7.455 | 5.848 | 7.533 | > 4 base-pairs |

Italics if ATP site distance <6.8 Å; italics if h2i-ps1β distance <8 Å; bold and assigned **tight** if both distance thresholds are met. Underlined structures are CMG structures.

Residues of the Cα–Cα distance calculations for each subunit interface (see Figs. 2 and 3) all reside in the highly conserved 152-amino acid AAA+ core[6]. The residues for comparative structures were obtained by sequence alignment[16] and confirmed by inspection of structures. The comparative residues correspond to a conserved proline of the Walker-A P-loop (Sso-PfMCM P330), 4 residues prior to the conserved arginine of the arginine finger (Sso-PfMCM T824), a conserved leucine of the helix/horseshoe region of the h2i (Sso-PfMCM L361), and the conserved lysine of the ps1β hairpin (Sso-PfMCM K785). The residues of the distance calculations are explicitly provided below:

**Class1a, Class 2a, Class 2b (Sso-PfMCM)**

| ATP site | h2i-ps1β |
|---|---|
| D/P330-E/T824 | D/L361-E/K785 |
| C/P330-D/T824 | C/L361-D/K785 |
| B/P330-C/T824 | B/L361-C/K785 |
| A/P330-B/T824 | A/L361-B/K785 |
| F/P330-A/T824 | F/L361-A/K785 |
| E/P330-F/T824 | E/L361-F/K785 |

**7W1Y[99] (HsMcm2–7)**

| ATP site | h2i-ps1β |
|---|---|
| HsMcm7/P383-HsMcm3/S474 | HsMcm7/L414-HsMcm3/K435 |
| HsMcm4/P512-HsMcm7/A510 | HsMcm4/L543-HsMcm7/K471 |
| HsMcm6/P398-HsMcm4/T639 | HsMcm6/L429-HsMcm4/K600 |
| HsMcm2/P525-HsMcm6/P525 | HsMcm2/L556-HsMcm6/K486 |
| HsMcm5/P383-HsMcm2/P652 | HsMcm5/L414-HsMcm2/K613 |
| HsMcm3/P347-HsMcm5/T509 | HsMcm3/L378-HsMcm5/K471 |

**5BK4[37], 9GJW[89], 9E2X[98], 7PMN[96], 7PMK[96], 7Z13[71] (ScMcm2–7)**

| ATP site | h2i-ps1β |
|---|---|
| ScMcm7/P462-ScMcm3/S538 | ScMcm7/L493-ScMcm3/K499 |
| ScMcm4/P570-ScMcm7/K550 | ScMcm4/L601-ScMcm7/K550 |
| ScMcm6/P577-ScMcm4/P697 | ScMcm6/L608-ScMcm4/K658 |
| ScMcm2/P545-ScMcm6/P704 | ScMcm2/L576-ScMcm6/K665 |
| ScMcm5/P418-ScMcm2/P672 | ScMcm5/L449-ScMcm2/K633 |
| ScMcm3/P411-ScMcm5/T545 | ScMcm3/L442-ScMcm5/K506 |

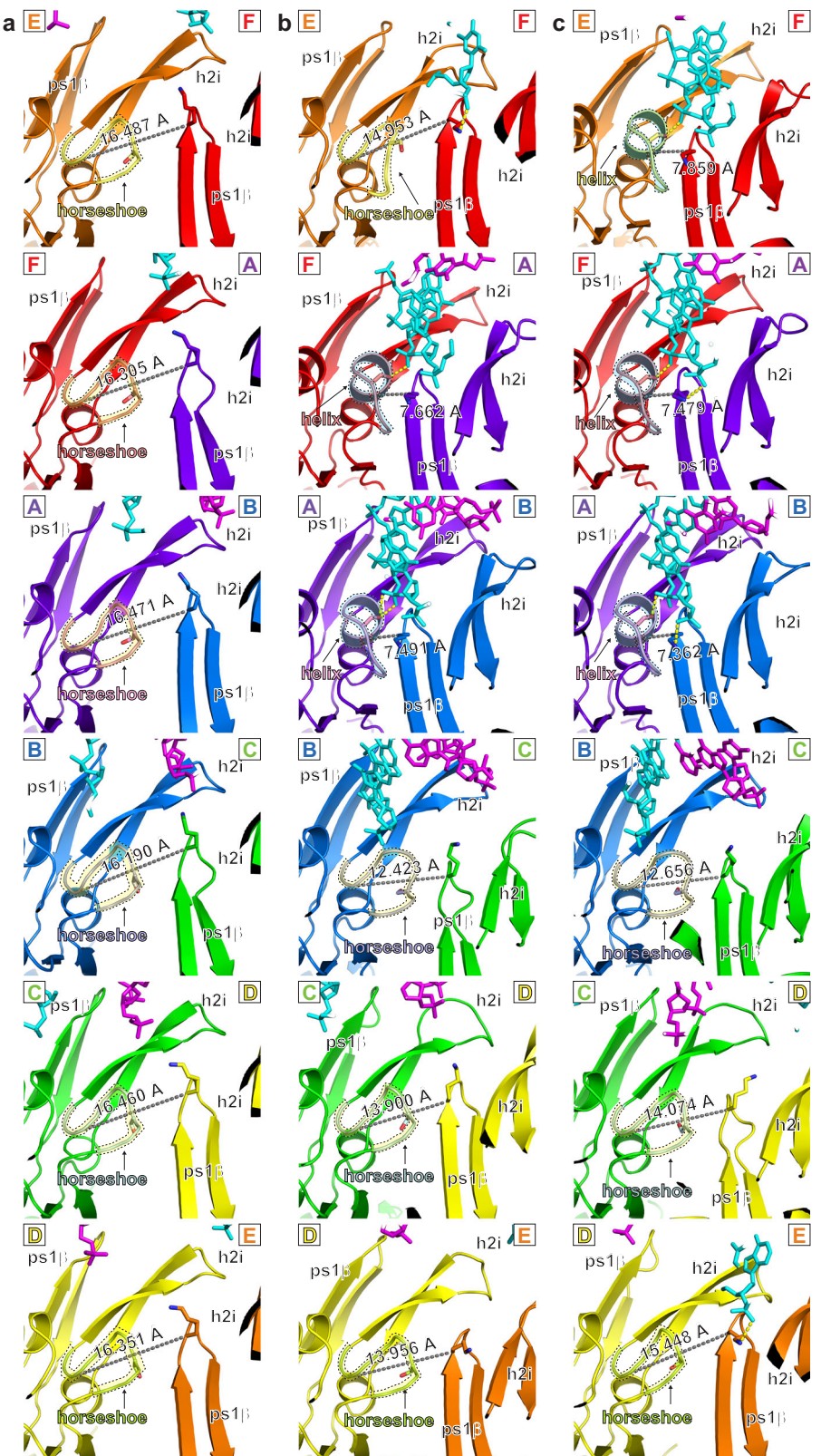

**Fig. 3 | Helix and horseshoe conformations of the MCM Helix-2-insert.** For clarity, each panel shows only two of the MCM subunits, and six panels depict all interfaces. DNA strands are depicted in cyan and magenta. The helix/horseshoe regions are emphasized in dashed outline with background shading. The conserved hydroxyl side-chain that interacts with a DNA phosphate in the helix conformation is shown in stick. The h2i L361 Cα - ps1β K785 Cα distance is illustrated in grey dash. An L361 Cα - K785 Cα distance <8 Å is one of the criteria used to assign a Tight interface (Table 1). **a** Class 1a has six horseshoe conformations with large intersubunit distances. **b** Class 2a has four horseshoe conformations with large intersubunit distances and two helix conformations with shorter intersubunit distances. **c** Class 2b has three horseshoe conformations with large intersubunit distances and three helix conformations with shorter intersubunit distances. Chain identifiers of the coordinate file are provided for each panel (chain A: purple, chain B: blue; chain C: green; chain D: yellow; chain E: orange; chain F: red) in a color scheme consistent with Fig. 1.

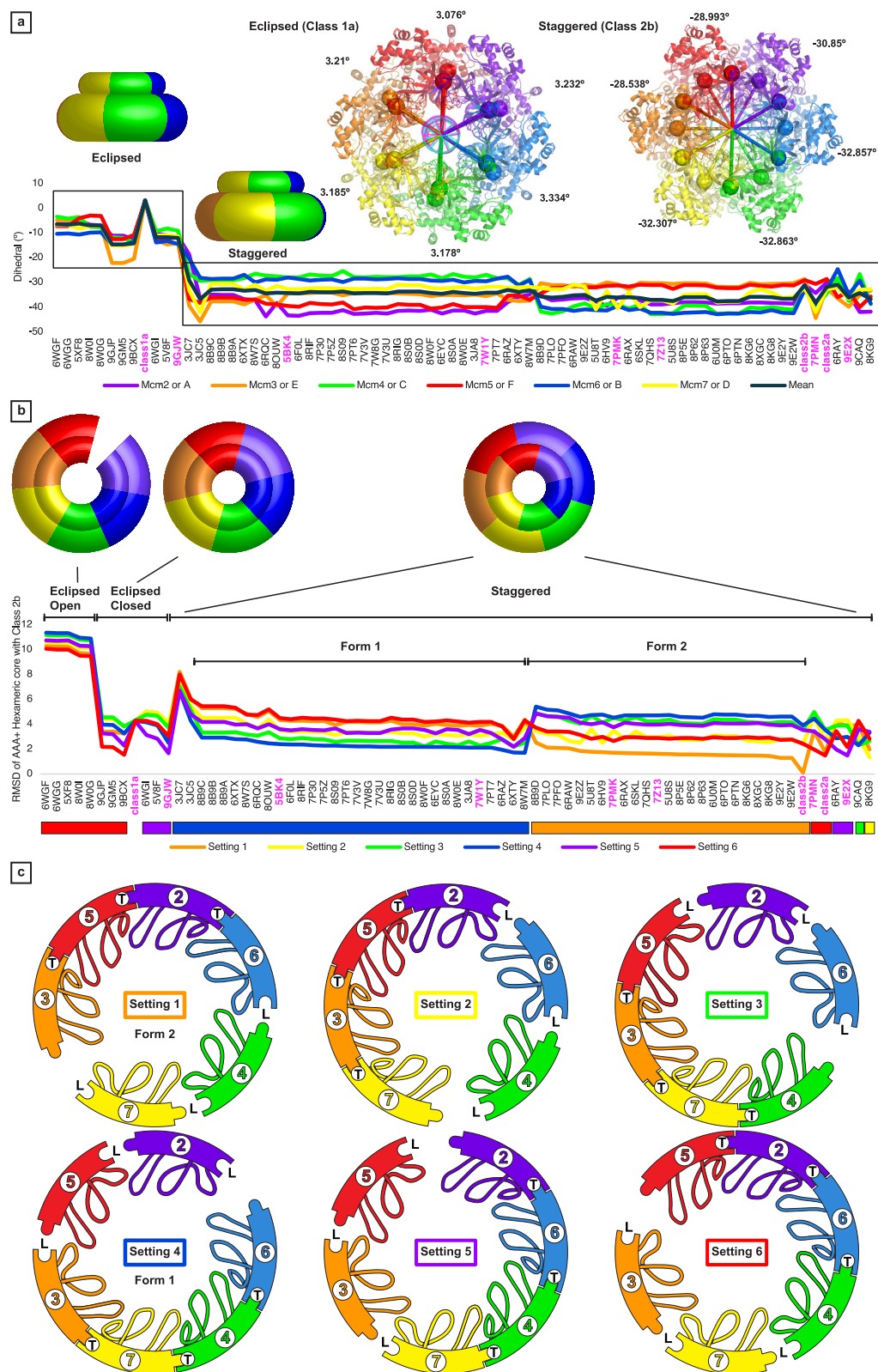

−38). These values are similar to those of Class 1a and Class 2, and so we will group them in the same eclipsed or staggered categories, with 11 in an eclipsed conformation and the majority (61) in a staggered conformation. Consistent with the prior description of the molecular basis for Cdt1 to drive specific inter-tier orientations[83], 100% of the structures that contain Cdt1 (9GM5[89], 9GJP[89], 9BCX[90], 9GJW[89], 6WGI[91], 5V8F[92], 6WGG[91], 5XF8[93]) adopt an eclipsed conformation in the

dihedral analysis. Only three structures that lack Cdt1 adopt an eclipsed conformation (8W0G[94], 8W0I[94], 6WGF[91]), and these structures all possess a contiguous opening at the Mcm2/5 interface across both tiers. Considered with the role of Cdt1 in generating an open Mcm2−7 ring[93], all eclipsed structures are either functionally prone to open across both tiers due to the presence of Cdt1, or the ring shows a bona fide continuous opening across both tiers. In contrast, 100% of CMG

**Fig. 4 | Comparison of the MCM:DNA structures with eukaryotic MCM structures clearly illustrates distinct fundamental architecture groups. a** Eclipsed and Staggered conformations of the two tiers are quantitatively defined by dihedral angles for each subunit using specific Cα positions of the N-tier OB-fold and Ctier AAA+ core (large spheres: *Sso*MCM P201 Cα, *Pf*MCM Y755 Cα) and the center of each tier. A plot of the dihedral angles for each eukaryotic Mcm2–7 structure shows a major difference between the staggered and eclipsed forms and that all subunits collectively belong to the same category for a given structure. Among the staggered complexes, two different major groups are present: one with Mcm4 and 6 dihedrals systematically above the mean and the other with Mcm4 and 6 dihedrals systematically below the mean. **b** A plot illustrates the RMSD of the collective hexamer of six 152-amino acid AAA+ cores against each of the six permutations (setting) of the Class2b AAA+ core hexamer. The eclipsed form has two prominently different RMSD categories that correspond to open-ring and closed ring structures. The staggered conformation has two obvious forms with the RMSD values of all settings in unison (Staggered Form I and Staggered Form II). The two plotted values together generate a consistent structure order for the two plots, where structures are binned eclipsed or staggered (eclipsed mean dihedral > −15°; staggered mean dihedral < −25°), sub-binned by lowest RMSD setting, and reverse sorted by the lowest RMSD value. Structures selected for illustration in the manuscript are highlighted in magenta. **c** Illustration of the configurations of Mcm2–7 that would provide a strong fit to each setting of Class 2b. Individual subunits are depicted with a unique color and labeled by subunit type (Mcm2: purple 2, Mcm6: blue 6; Mcm4: green 4; Mcm7: yellow 7; Mcm3: orange 3; Mcm5: red 5). The two hairpins of each subunit are depicted with the ps1β as a single loop and the h2i including a feature that is either a helix or horseshoe. Tight and loose interfaces are denoted T and L, respectively. Notably, the arrangements of Setting 1 and Setting 4, which comprise nearly all staggered structures (see (**b**)), are offset by 180°.

structures in the analysis are in a staggered conformation and closed ring. The overall analysis indicates that Cdt1 drives an eclipsed conformation while Cdc45 and GINS drive a staggered conformation. The Class 1a structure is unique within the eclipsed category because it consists solely of MCM proteins in a closed ring. Class 1a also has dihedral values that are significantly more positive than all other structures analyzed (Fig. 4a). The dihedral values of the Class 2 structures are very similar to those of eukaryotic staggered complexes (Fig. 4). The gross similarities of Class 1a to several eukaryotic eclipsed structures (5XF8[93], 6WGG[91], 5V8F[92], 9GJW[89], and 9BCX[90]) and Class 2b to a eukaryotic staggered structure (PDB 7Z13[71]) is illustrated with a molecular morph in Supplementary Movie 1. Our subsequent analysis will focus on the ATPase C-tier and ignore the N-tier.

## Comparative analysis II: AAA+ hexamer core RMSD values define two principal forms of staggered eukaryotic Mcm2–7

We also quantitatively compared MCM hexamer structures based on the collective architecture of all six ATPase domains. Analogous to the highly specific inter-subunit arrangement for catalytically competent F1-ATPase[22], a highly specific MCM inter-subunit molecular architecture is required to generate a tight MCM ATPase site poised for hydrolysis. We have previously shown that MCM proteins have an ATPase core that is nearly invariantly 152-amino acids in length and highly conserved in sequence and structure[16]. We have used both properties to identify the eukaryotic Mcm2–7 structures that most similarly resemble our structures for overall distribution of tight and loose ATPase sites. To identify the most appropriate comparative eukaryotic MCM structures for the collective arrangement of the six MCM ATPase cores as a unit, RMSDs of the hexameric ATPase core of all eukaryotic Mcm2–7 structures with six AAA+ domains of the PDB (77 eligible structures, including all 72 of the dihedral analysis) were calculated[86,95] against all six permutations of the Class 1a structure and against all six permutations of the Class 2b structure (462 comparisons each, see Supplementary Fig. 12). Each of the six permutations will be referred to as a specific setting of the complex. The top 30 RMSD alignments (Supplementary Tables 7 and 8) were visually inspected. The Class1 structures did not have significant matches (Supplementary Table 7), suggesting that no eukaryotic Mcm2–7 structures with six loose interfaces are represented in the PDB. In contrast, the Class 2 structures provided several strong matches (Supplementary Table 8).

Within the RMSD analysis, two principal forms of staggered Mcm2–7 structures are obvious. The 72 Mcm2–7 structures eligible for both the dihedral and RMSD analysis (above) were binned to eclipsed (11 structures) or staggered (61 structures) categories, sub-binned by the setting of Class 2b that provided the lowest RMSD, and each bin was reverse-sorted by the RMSD value of that setting (Fig. 4). The staggered conformation has two obvious forms with the RMSD values of all settings in unison. These two forms comprise nearly all staggered Mcm2–7 complexes (56 of 61 structures) and will be referred to as Staggered Form I (32 structures) and Staggered Form II (24 structures)

with Form I best fitting Class 2b in setting 4 and Form II best fitting Class 2b in setting 1. The dihedral analysis similarly identifies two principal forms that include the same respective structures (Fig. 4a). Form I systematically has Mcm4 and Mcm6 dihedrals above the mean; Mcm3 and Mcm7 dihedrals near the mean; and Mcm2 and Mcm5 dihedrals below the mean. Form II has Mcm3 and Mcm5 dihedrals above the mean; Mcm2 and Mcm7 dihedrals near the mean; and Mcm4 and Mcm6 dihedrals below the mean. Notably, settings 1 and 4 that define these two principal forms are offset by 180° (Fig. 4c) and thus appear approximately opposite one another, suggesting they could fulfill opposing functions.

We associate Form I with stable association with fully duplex DNA and Form II with DNA melting. Form I structures exist within multiple species, including single-hexamers, double-hexamers, and CMG. Some Form I structures encircle DNA, either dsDNA or ssDNA, while others do not contain DNA. Importantly, all Form I structures that encircle dsDNA have fully base-paired DNA at the ATPase tier. All Form II structures (24 total) are CMG structures. Of these, 3 have melted DNA bound by the ATPase tier hairpins (PDB 7PMK[96], 7Z13[71], and 7QHS[71], see below), and 19 have ssDNA bound by the hairpins. Two structures are low-resolution structures of complexes that included ssDNA during sample preparation but did not include DNA in the structural models (PDB 6PTN[97]: 5.80 Å, PDB 6PTO[97]: 7.00 Å). These two structures may have ssDNA bound at the ATPase tier hairpins that was not readily discerned at the reported resolution. If so, then all 24 Form II structures consist of CMG molecules with ssDNA regions bound by the ATPase hairpins.

Based on inspection of the top 30 RMSD fitting structures of Class 2b (Supplementary Table 8), we selected PDB 7PMN[96], 7PMK[96], 7Z13[71], 9E2X[98], and 9GJW[89] for more thorough comparison and analysis below. None of these comparative structures belong to staggered Form I. With no strong RMSD fits to Class 1a, we selected two staggered Form I Eukaryotic MCM double-hexamer structures encircling DNA that is fully duplexed at the ATPase tier (PDB 5BK4[37] and PDB 7W1Y[99]) as comparative cases for the Class 1 structures. The tight versus loose criteria outlined in the section above was applied to these selected structures (Table 1). The structures and their comparative equivalents are analyzed in finer detail below.

## Fundamental class 1: eclipsed hexamer; fully base-paired DNA at both tiers

The first fundamental class places the two tiers of the MCM hexameric ring in an eclipsed conformation with both tiers encircling a contiguous dsDNA molecule that appears undistorted and fully base-paired. The central channel is more dilated than in the other fundamental hexameric class (see below), and the DNA-binding hairpins of the ATPase domain only marginally interact with the encircled DNA (Fig. 5 and Supplementary Figs. 13 and 14). For each structure of Fundamental Class 1, all h2i are in the horseshoe conformation (Fig. 3 and Supplementary Figs. 15–17), likely contributing to the limited direct binding between the h2i hairpin and the DNA. All Class 1 structures,

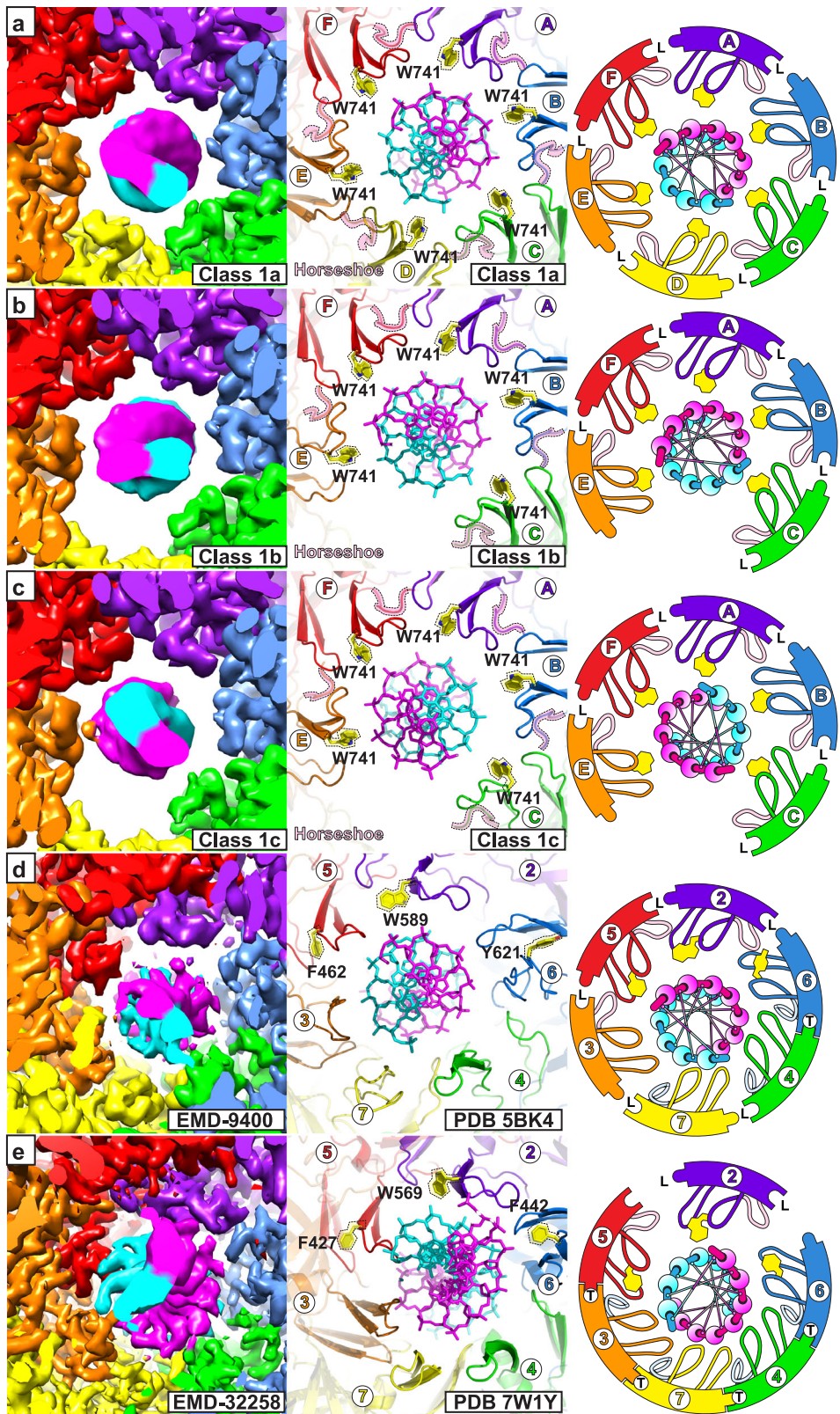

therefore, have loose interfaces, consistent with nucleotide-free states, as required as a first step towards replication initiation[38]. Within Fundamental Class 1, three subclasses are identified with different numbers of well-ordered AAA+ ATPase subunits.

Class 1a has six well-defined AAA+ domains at the C-tier and a well-defined B-form DNA duplex (Fig. 1a and Supplementary Figs. 10a and 11a). Ignoring the encircled DNA, the protein complex appears sixfold symmetric such that six permutations are indistinguishable. As such, the origin of the DNA helical axis defines the specific MCM:DNA permutation of each cryo-EM particle. The hairpins of the ATPase domain do not closely contact the DNA (Fig. 5a and Supplementary Figs. 13a and 14a). All six h2i are in the horseshoe conformation with corresponding loose subunit interfaces (Supplementary Figs. 15a–17a).

**Fig. 5 | Comparison of DNA-binding at the ATPase tier by Sso-PfMCM:DNA Fundamental Class 1 with Mcm2–7 Staggered Form I structures that encircle fully duplex DNA.** Images are cutaway at a depth that emphasizes the two DNA-binding hairpins of the ATPase domain. For the Class 1 structures (a–c), the hairpins do not intimately contact either DNA strand, and each h2i hairpin adopts a horseshoe conformation (highlighted with dashed outline and magenta background shade in middle panels) and has a loose intersubunit interface (L in right panel cartoons). The h2i W741 aromatic groups (highlighted with dashed outline and yellow background shade in middle panels) are distant from the DNA. Unsharpened maps are illustrated to emphasize the general MCM:DNA regions. Structures of Eukaryotic Mcm2–7 double-hexamers encircling double-stranded DNA (d PDB 5BK4[37]; EMDB-9400[37] and e PDB 7W1Y[99]; EMDB-32258[99]), both belonging to Staggered Form I (see Fig. 4), similarly do not place the aromatic groups of their h2i (highlighted with dashed outline and yellow background shade in middle panels) near the DNA. Structure chain identifiers are provided for each model in the middle and right panels (Class 1 chains A–F in (a–c) and Mcm2: purple 2, Mcm6: blue 6; Mcm4: green 4; Mcm7: yellow 7; Mcm3: orange 3; Mcm5: red 5). Right panel cartoons depict the two hairpins as in Fig. 4c with additional coloring for the h2i as light blue for helix or light pink for horseshoe. H2i aromatic groups are shown in yellow for right panel cartoons.

Class 1b has five well-defined AAA+ domains at the C-tier and a B-form DNA duplex (Fig. 1b and Supplementary Figs. 10b and 11b). This protein complex is not $C_6$-symmetric due to the poor order of one of the AAA+ domains. The arrangement of five ordered ATPase domains defines a specific permutation of MCM hexamer particles. Two different DNA modes are observed for this asymmetric species. In both cases, the h2i of the orange subunit located clockwise to the disordered ATPase subunit (in the Top view from the N-terminal side of Fig. 1) makes the closest approach to a DNA strand. The polarity of the contacted strand differs with the cyan strand in closest proximity to the orange subunit h2i in this subclass. Although this h2i is the closest approach between the ATPase tier and the DNA, it does not appear to form specific interactions (Fig. 5b and Supplementary Figs. 13b and 14b). All five ordered ATPase domains are in the horseshoe conformation with corresponding loose subunit interfaces (Supplementary Figs. 15b and 16b).

Class 1c also has five well-defined AAA+ domains at the C-tier and a B-form DNA duplex (Fig. 1c and Supplementary Figs. 10c and 11c). This protein complex is also not $C_6$-symmetric due to the poor order of one of the AAA+ domains. The arrangement of five ordered ATPase domains defines the specific permutation of MCM hexamer particles and does not appear distinguishable from Class 1b based on the protein. In this subclass, the orange subunit located clockwise to the disordered ATPase subunit (in the Top view from the N-terminal side of Fig. 1) makes the closest approach to the magenta DNA strand, the opposite strand polarity as Class 1b. Although this h2i is the closest approach between the ATPase tier and the DNA, it does not appear to form specific interactions (Fig. 5c and Supplementary Figs. 13c and 14c). All five ordered ATPase domains are in the horseshoe conformation with corresponding loose subunit interfaces (Supplementary Figs. 15c and 16c).

The Class 1 structures are similar to those of the eukaryotic pre-melting complexes within the yeast Mcm2–7 double hexamer (PDB 5BK4[37], Fig. 5d) and the human Mcm2–7 double hexamer (PDB 7W1Y[99], Fig. 5e) because the DNA is fully duplex at the C-tier, and the aromatic groups of the h2i hairpin (yellow and highlighted with dashed outline) are distant from the DNA. While all Class 1 structure subunits are in the horseshoe conformation with corresponding loose interfaces, the structure of PDB 5BK4[37] has three horseshoe (Mcm3, 5, and 2) and three helix conformations (Mcm6, 4, and 7; Supplementary Fig. 17e). The structure of PDB 7W1Y[99] only differs from the PDB 5BK4[37] helix/horseshoe arrangement at Mcm3, which appears to be horseshoe in PDB 5BK4[37] and helix in of PDB 7W1Y[99]. The transformation of an MCM ATPase tier structure within the Mcm2–7 double hexamer (PDB 5BK4[37] or PDB 7W1Y[99]) from a species with some tight interfaces to 0-tight/6-loose interfaces in Class 1a provides a representation of the initial exhaust of ADP from the MCM double-hexamer that is an essential replication initiation early step[38].

**Fundamental class 2: staggered hexamer; DNA partially opened and exclusively at C-tier**
The second fundamental class approximately staggers the domains that comprise each tier (Fig. 1 and Supplementary Figs. 10 and 11). This inter-tier rotation is similar to that observed in the crystal structure of an archaeal MCM hexamer in complex with single-stranded DNA[31]. It is also similar to the orientations of eukaryotic CMG[41,71,72,96–98,100–110] and for each MCM hexamer of the eukaryotic Mcm2–7 double-hexamer[30,37,111–114] (see Fig. 4). In contrast to Fundamental Class 1, Fundamental Class 2 has extensive interactions between the C-tier DNA-binding hairpins and one DNA strand in a binding mode that is highly similar to that with single-stranded DNA[31]. Different numbers of h2i helix/horseshoe conformations (Fig. 3 and Supplementary Figs. 15–17) with concurrent differences in the numbers of bound nucleotides define two subclasses. These differences suggest successive steps in a melting mechanism that will be elaborated in a subsequent section.

In Class 2a, the h2i and ps1β hairpins of three adjacent subunits form a spiral staircase that binds 5 consecutive nucleotides of one DNA strand (cyan strand, Fig. 1d and Supplementary Figs. 10d and 11d). Two nucleotides of the opposing strand (magenta) are disordered. These nucleotides would complement the first two cyan strand nucleotides that are tightly bound by the MCM hairpins. The loss of these base-pair interactions (Fig. 6a and Supplementary Figs. 13d and 14d) indicates that this region of DNA has melted and that this melting is driven by interaction of the cyan strand with the MCM hairpins. The binding mode with the cyan strand is analogous to the 6-subunit spiral staircase that the MCM ring uses to bind 10 consecutive nucleotides of ssDNA[16,31]. To bind consecutive nucleotides in this mode, participating subunits must adopt the helix h2i conformation for two reasons: (1) to position the conserved hydroxyl group where it can bind a DNA phosphate group; and (2) to develop a tight subunit interface such that adjacent subunits are positioned to bind DNA in the same mode. The configuration of Class 2a with two consecutive helix conformations (Supplementary Figs. 15d–17b) enables three consecutive subunits to engage in binding the cyan strand (Fig. 6a and Supplementary Figs. 13d and 14d). This structural class has two tight subunit interfaces and four loose subunit interfaces that are defined by the two helix conformations and four horseshoe conformations of the h2i. This structure is highly similar to a eukaryotic CMG structure that also melts 2 DNA base-pairs (PDB 7PMN[96], Fig. 6b). The two nucleotides of unpaired DNA in the structure of PDB 7PMN[96] represent bona fide DNA melting because the DNA substrate does not possess an unpaired 5′-tail. As with the Class 2a structure, the two nucleotides of the complementary strand are disordered. Notably, both structures stack an h2i aromatic side-chain on the sugars and bases of DNA nucleotides that are melted (yellow and highlighted in dashed outline in Fig. 6).

Class 2b has four subunits engaged in binding DNA (Fig. 1e and Supplementary Figs. 10e and 11e). This class is analogous to Class 2a with three consecutive subunits in the helix conformation (Supplementary Figs. 15e–17c), enabling four consecutive subunits to bind the 7 nucleotides of the cyan strand (Fig. 5c and Supplementary Figs. 13e and 14e). Four nucleotides of the opposing strand (magenta) are disordered. These nucleotides complement the first four cyan strand nucleotides that are tightly bound by the hairpins. The loss of these base-pair interactions (Fig. 6c and Supplementary Figs. 13e and 14e) indicates that this region of DNA has melted and increased DNA melting by two nucleotides relative to Class 2a. Class 2b has three tight subunit interfaces and three loose subunit interfaces (Fig. 2) with three helix conformations and three horseshoe

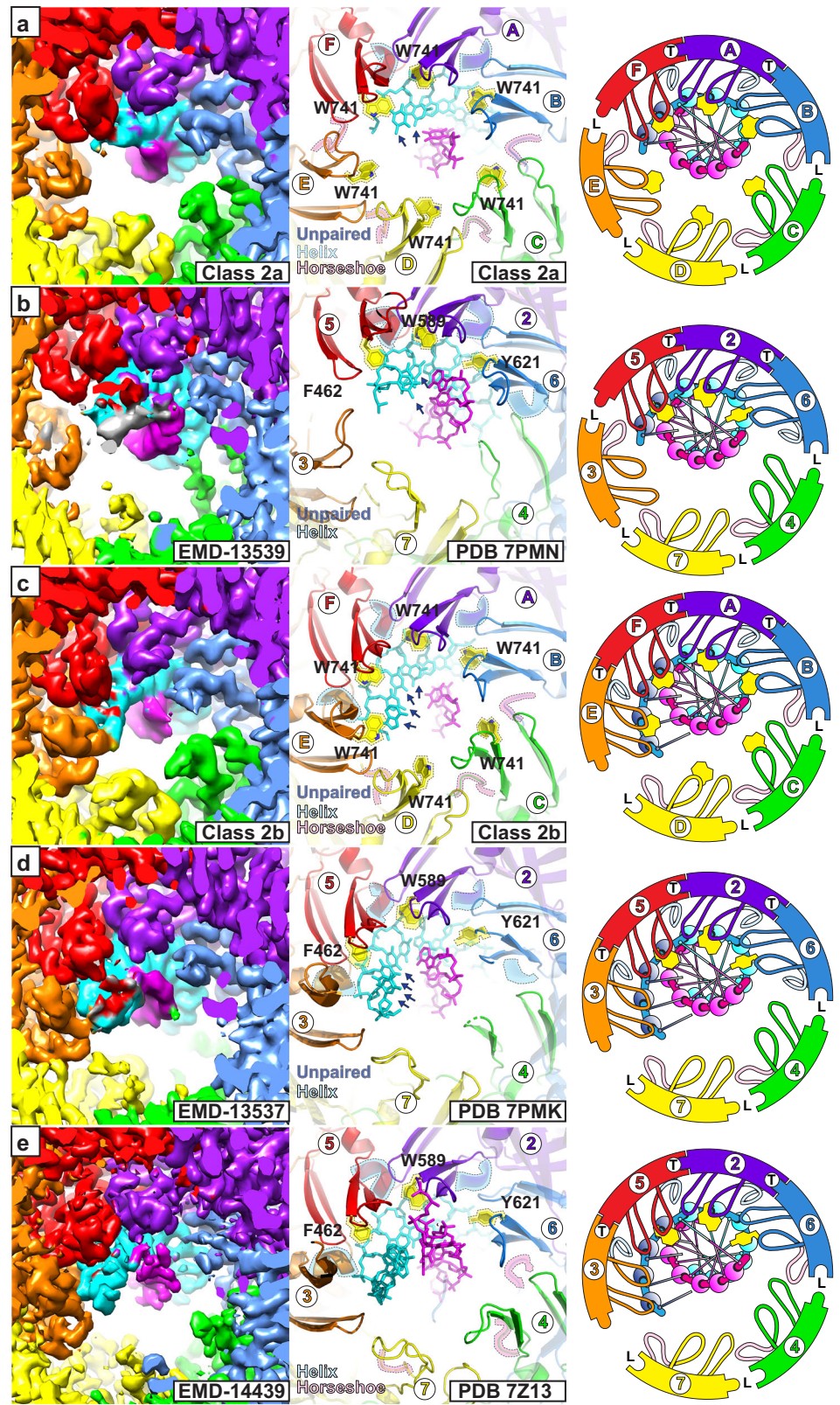

conformations of the h2i (Fig. 3). Class 2b is highly similar to two different eukaryotic CMG structures with melted DNA (PDB 7PMK[96] and PDB 7Z13[71]), suggesting a universal role for this structural form. Specifically, these structures adopt 3-helix/3-horseshoe conformations and similarly bind the DNA strand depicted in cyan with analogous hairpin-DNA interactions (Fig. 6d, e and Supplementary Fig. 17d). The four nucleotides of unpaired DNA in the structure of PDB 7PMK[96]

represent bona fide DNA melting because the DNA substrate does not possess an unpaired 5′-tail. As with the Class 2b structure, the four nucleotides of the complementary strand are disordered. The structures consistently stack an h2i aromatic side-chain on the sugars and bases of the bound DNA strand that is melted (yellow and highlighted in dashed outline in Fig. 6). The helix/horseshoe distribution of PDB 7Z13[71] is opposite that of PDB 5BK4[37] (Supplementary Fig. 17) and

**Fig. 6 | Comparison of DNA-binding at the ATPase tier among forms that encircle melted DNA.** Images are cutaway at a depth that emphasizes the two DNA-binding hairpins of the ATPase domain. For the Class 2 structures (**a**, **c**), subunits where the h2i hairpins adopt a helix conformation (highlighted in dashed outline with blue background shade in middle panels) intimately associate with the cyan DNA strand and create a tight intersubunit interface (T in right panel cartoon). Two base-pairs are melted in Class 2a, and four are melted in Class 2b (dark blue spheres in right panels). The h2i W741 aromatic groups (highlighted with dashed outline and yellow background shade in middle panels) stack on the sugars of melted DNA nucleotides. **b** Class 2a is highly similar to the eukaryotic MCM:DNA Structure PDB 7PMN[96] (EMDB-13539[96]) with two tight interfaces and two base-pairs melted. **d** Class 2b is highly similar to the eukaryotic MCM:DNA Structure PDB 7PMK[96] (EMDB-13537[96]) with three tight interfaces and four base-pairs melted. **e** Class 2b is highly similar to the eukaryotic MCM species encircling melted DNA (PDB 7Z13[71]; EMDB-14439[71]), where three h2i hairpins adopt a helix conformation and similarly associate with the cyan DNA strand. The eukaryotic comparative structures are all CMG structures that belong to Staggered Form II (see Fig. 4) with h2i aromatic residues (Mcm5 F462, Mcm2 W589, and Mcm6 Y621) all stacked on the sugars of melted DNA nucleotides as in the Class 2 structures. Structure chain identifiers are provided for each model in the middle and right panels (Class 2 chains A-F in (**a**, **c**) and Mcm2: purple 2, Mcm6: blue 6; Mcm4: green 4; Mcm7: yellow 7; Mcm3: orange 3; Mcm5: red 5). Right panel cartoons depict the two hairpins as in Fig. 4c with additional coloring for the h2i as light blue for helix or light pink for horseshoe. H2i aromatic groups are shown in yellow for right panel cartoons.

reflects the opposite configurations of eukaryotic staggered Form I and eukaryotic staggered Form II (Fig. 4).

## Fundamental class 3: staggered heptamer; fully base-paired DNA at both tiers

A third fundamental class consists of a heptameric MCM ring encircling fully duplexed DNA. (Supplementary Fig. 18). Ignoring the encircled DNA, the protein complex appears sevenfold symmetric such that seven permutations are indistinguishable. As such, the origin of the DNA helical axis sets the specific MCM:DNA permutation of each cryo-EM particle. The two tiers are in a staggered conformation and not eclipsed. This is a minor class and is not predicted to function biologically. It is presented here out of completeness and not discussed further.

## DNA-interactions

Both fundamental hexameric ring structural classes encircle DNA, but they interact with DNA differently. While Fundamental Class 1 has limited direct interaction with the DNA, Fundamental Class 2 strongly interacts with the DNA sugar-phosphate backbone, principally with one strand (depicted in cyan in Figs. 1 and 6). For a given subunit, four total polar interactions of the two hairpins with the cyan strand match those observed for ssDNA-binding by the corresponding residues of a related archaeal MCM hexamer (PDB 6MII, Fig. 7a). These interactions are between phosphate groups of the DNA backbone and atoms that are fully conserved in all MCM proteins and are therefore feasible for all eukaryotic and archaeal MCM subunits. These interacting atoms reside on the h2i and ps1β DNA-binding hairpins of the ATPase domain. The h2i is visually more prominent than the ps1β because it extends further into the central channel. This hairpin forms two polar interactions with the cyan strand DNA backbone. For DNA-interacting h2i, the conserved hydroxyl side-chain of S357 and main-chain amide of A732 interact with the phosphates of consecutive nucleotides of DNA. Notably, S357 is within the residues that comprise the helix/horseshoe conformations, and the helix conformation uniquely places it in position to interact with the DNA backbone. For DNA-interacting ps1β, the conserved positively charged side-chain of K785 and the main-chain amide of A786 interact with adjacent DNA phosphate groups. The polar interactions of the two hairpins are shown in magenta (Fig. 7b). Mutation of the ps1β lysine completely disables the strand displacement activity of MCM[31,115]. Mutation of the h2i hydroxyl group in SsoMCM (T369A) greatly reduces the cooperativity of unwinding[31]. SsoMCM T369A mutants were not well-behaved, and the only construct successfully isolated had a specialized internal loop deletion[31]. The reduced unwinding cooperativity and construct behavior are consistent with a role for this residue to interact with a neighboring subunit to develop a tight interface.

The W741 aromatic group stacks on the sugar of the DNA sugar-phosphate backbone (yellow, Fig. 7b). This stacking interaction is conserved in the ssDNA bound structure of SsoMCM (Y386, Fig. 7a). This stacking interaction is not needed to unwind DNA in a strand displacement assay based on alanine mutagenesis with SsoMCM[31], and

we have previously suggested that the stacking interaction could be important for DNA melting. Notably, the SsoMCM Y386A mutation eliminates the aromatic group from all 6 subunits of the SsoMCM homo-hexamer.

## Aromatic wedge is conserved in archaeal MCM; eukaryotic Mcm5, 2, 6; SV40 large T-antigen and papillomavirus E1

As illustrated above, the DNA-interacting residues of the MCM ATPase hairpins have two categories: (1) polar residues (K785 and S357) that interact with the DNA backbone and are essential for DNA unwinding; and (2) aromatic residues (W741) that stack on DNA sugars and bases and are dispensable for DNA-unwinding. The separate functions of these residues strongly resemble the separate functions of two residues of the papillomavirus E1 ps1β, where the analogous lysine (K506) is essential for unwinding DNA, while the aromatic histidine (H507) is dispensable for unwinding DNA[70]. Instead, H507 has a critical role in DNA replication, DNA melting, and the functional progression from a double-trimer to a double-hexamer[70]. Of the 8 H507 mutants tested (A, V, L, R, M, N, F, and Y), only H507F supported in vivo replication in a transient DNA replication assay[70], and only the aromatic mutants (H507F and H507Y) supported in vitro replication in a cell-free replication system[70]. Based on permanganate reactivity, only the H507F mutant melted DNA in the T-rich region that is anticipated to be located at the E1 ATPase tier[70], and only the H507F mutant provided unwinding in an origin unwinding assay[70].

We have previously noted that the SsoMCM h2i aromatic residue Y386 stacks on DNA sugars and bases very similar to the E1 ps1β aromatic H507 residue and suggested these residues may serve similar functions for melting and replication initiation[31]. We now strengthen this analogy based on a recent structure of SV40 L-tag, a close relative of E1, bound to melted DNA (PDB 9KAE[27], Fig. 7c). This structure shows five unpaired nucleotides of DNA that represent bona fide DNA melting because the substrate did not have an exposed single-stranded region. In this structure, the aromatic histidine side-chain H513 stacks on the sugars and bases of the melted DNA with strong similarity to the aromatic residue of archaeal MCM (our Class 2 structures) and eukaryotic Mcm5, 2, and 6 (PDB 7PMN[96], PDB 7PMK[96], PDB 7Z13[71], and PDB 7QHS[71]). The nucleotides not bound by L-Tag that have lost base-pairing interactions are disordered, analogous to the behavior of the melted DNA of our Class 2 structures and the eukaryotic CMG structures of PDB 7PMN[96] and 7PMK[96] (Fig. 6). Interestingly, although the central channel hairpins bind one DNA strand with identical polarity, the intact double-stranded region is observed at the N-tier for L-tag (PDB 9KAE[27]) and at the C-tier for the MCM structures (Class 2a, Class 2b, 7PMN[96], 7PMK[96], 7Z13[71], and 7QHS[71]). Similar to MCM and E1, residues of the L-tag ps1β have different functions. Lysine K512 is essential for unwinding DNA in a strand displacement assay[116], but DNA unwinding in a strand displacement assay is less sensitive to H513 mutation (6 of 10 H513 mutants retain activity)[116]. Based on these factors, we will refer to these aromatic residues as constituting an aromatic wedge to melt DNA for each complex.

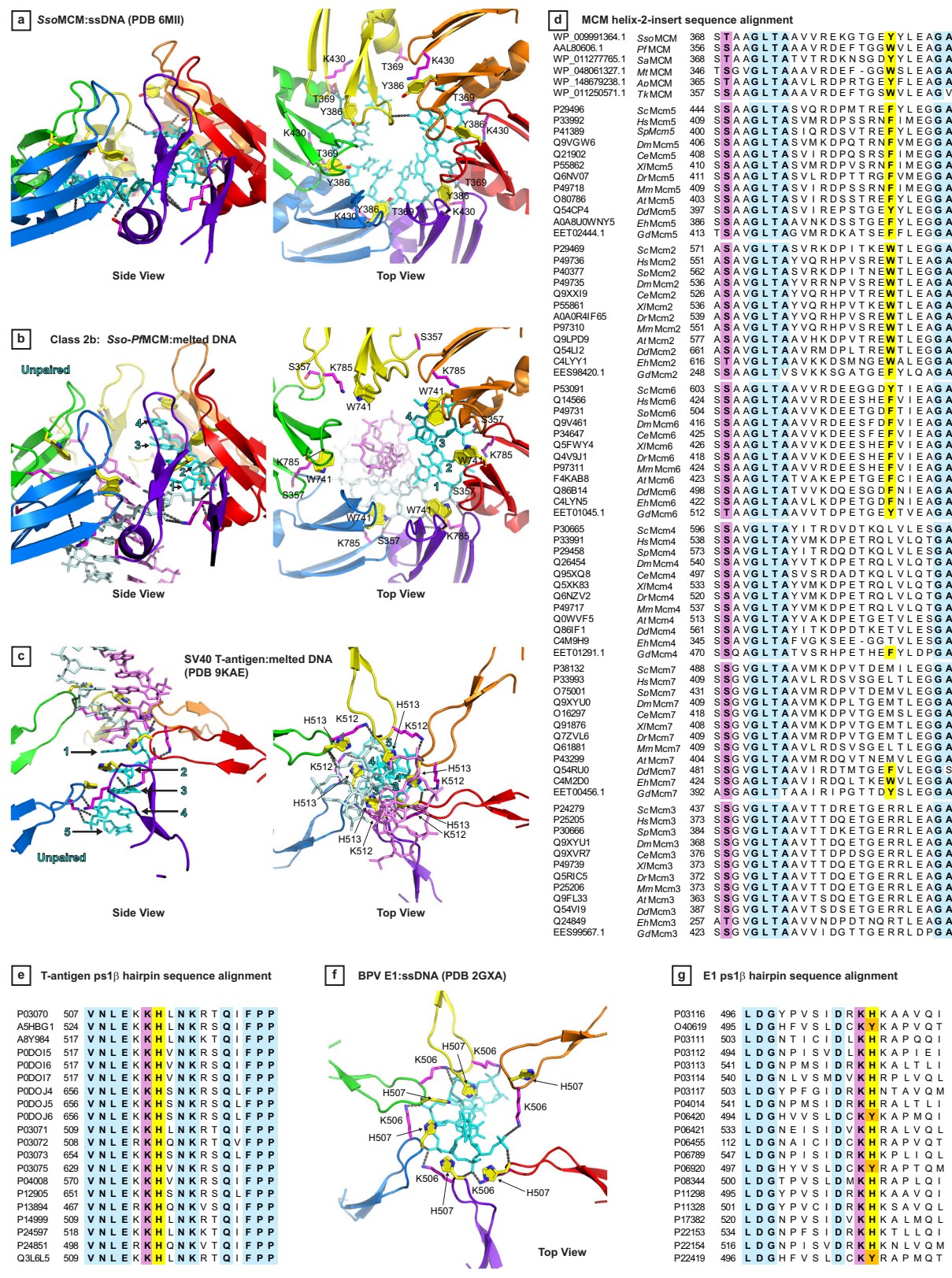

Aromatic wedge residues are inherently present at all six subunits of archaeal homohexamers, but they follow a striking conservation pattern in eukaryotic Mcm2–7 that matches the arrangements of eukaryotic staggered Form I and staggered Form II (Fig. 4). Three consecutive subunits in the ring, Mcm5, Mcm2, and Mcm6, conserve an aromatic residue on the h2i at the same residue position as archaeal MCM (Fig. 7d). These aromatic groups stack on DNA sugars and bases

in complexes with melted DNA (Fig. 6) but are distant from DNA in complexes with unmelted dsDNA (Fig. 5). In contrast, the other three subunits, Mcm4, Mcm7, and Mcm3, are nearly always not aromatic (Fig. 7d). These two categories of h2i groups (aromatic: Mcm5, 2, 6 vs. non-aromatic Mcm4, 7, 3) correlate with the arginine finger side of tight interfaces for the two eukaryotic staggered forms (Fig. 4). Specifically, Mcm4, 7, and 3 are the subunits at the arginine finger side of

**Fig. 7 | DNA-binding residues and an aromatic wedge are conserved in archaea, eukaryotes, and DNA tumor viruses. a** The ATPase tier of *SSo*MCM encircling ssDNA (PDB 6MII[31]) binds DNA with ps1β K430 (magenta) and h2i T369 (magenta). Only the ps1β and h2i hairpins of the protein are shown. The h2i aromatic Y386 (yellow) stacks on the DNA sugars. Polar interactions are indicated with grey dashes. **b** *Sso-Pf*MCM Class 2b binds one strand of dsDNA with ps1β K785 (magenta) and h2i S357 (magenta). Only the ps1β and h2i hairpins of the protein are shown. The h2i aromatic W741 (yellow) stacks on the DNA sugars. Polar interactions are indicated with grey dashes. Four base-pairs are melted in the vicinity of the W741 aromatic wedge (indicated by cyan numbers). **c** SV40 Large T-antigen bound to melted DNA (PDB 9KAE[27]) binds one strand of DNA with ps1β K512 (magenta), and the ps1β aromatic H513 (yellow) stacks on DNA sugars with striking similarity to the MCM h2i aromatic wedge. Only the ps1β hairpins of the protein are shown. Five base-pairs are melted in the vicinity of the H513 aromatic wedge (indicated by cyan numbers). Polar interactions are indicated with grey dashes. **d** The hydroxyl group of the MCM h2i that binds DNA is universally conserved in all MCM subunits while the h2i aromatic residues are specifically conserved at Mcm5, 2, and 6. Residues with 95% conservation are in bold with cyan shading. The universally conserved h2i hydroxyl group that binds DNA is in bold and shaded magenta. The h2i aromatic residue position is in bold and shaded yellow when aromatic. The alignment has archaeal sequences followed by eukaryotic sequences grouped by subunit type and placed in sequential order about the ring. Archaeal MCM: *Sso Saccharolobus sol-fataricus, Pf Pyrococcus furiosus, Sa Sulfolobus acidocaldarius, Mt*

*Methanothermobacter thermautotrophicus, Ap Aeropyrum pernix, Tk Thermococcus kodakarensis*. Eukaryotic Mcm2–7: *Sc Saccharomyces cerevisiae, Hs Homo sapiens, Sp Schizosaccharomyces pombe, Dm Drosophila melanogaster, Ce Caenorhabditis elegans, Xl Xenopus laevis, Dr Danio rerio, Mm Mus musculus, At Arabidopsis thaliana, Dd Dictyostelium discoideum, Eh Entamoeba histolytica* ATCC 30459, *Gd Giardia duodenalis* ATCC 50581. For *Pf*MCM, residues 362–728 have been removed to account for the intein removed from the mature protein. For *Tk*MCM, residues 363-502 have been removed to account for an intein removed from the mature protein. **e** The SV40 Large T-antigen ps1β DNA-binding lysine (in bold and shaded magenta) and aromatic wedge (in bold and shaded yellow) are universally conserved. All 20 Prosite[137] reviewed sequences are shown. Residues with 95% conservation are in bold with cyan shading. **f** The structure of papillomavirus E1 bound to ssDNA (PDB 2GXA[23]) binds one strand of DNA with ps1β K506 (magenta), and the ps1β aromatic H507 (yellow) stacks on DNA sugars equivalently to SV40 T-antigen and similar to the MCM h2i aromatic. Polar interactions are indicated with grey dashes. **g** The E1 ps1β DNA-binding lysine (in bold and shaded magenta) is universally conserved, and the aromatic group is conserved as histidine (in bold and shaded yellow) or tyrosine (in bold and shaded orange). Of the 80 Prosite[137] reviewed sequences (21 shown), 71 are histidine and 9 are tyrosine. An alignment with all 80 sequences is provided in Supplementary Fig. 20. Residues with 95% conservation among the 80 sequences are in bold with cyan shading. Accession numbers are provided to the left of each sequence for each alignment panel.

three tight interfaces of Form I (Setting 4 of Fig. 4), and Mcm5, 2, and 6 are the subunits at the arginine finger side of three tight interfaces of Form II (Setting 1 of Fig. 4). Thus, the presence or absence of an aromatic wedge provides the underlying basis for Form I to adopt a stable structure that specifically does not melt DNA, and Form II to adopt an alternate stable structure that melts DNA. Analogous to the conserved ps1β of E1 and L-tag, all archaeal and eukaryotic Mcm2–7 conserve a lysine on the ps1β[29,31,115] (magenta, Supplementary Fig. 19) and conserve a hydroxyl group of the h2i[31] (magenta, Fig. 7d). These residues very likely are functionally equivalent to the conserved lysines of L-tag and E1 (Fig. 7e–g and Supplementary Fig. 20). Interaction of these residues with DNA is sufficient to unwind DNA in a strand displacement assay, but DNA melting requires the functionally distinct aromatic residues of the hairpins.

**Mechanism for sequential melting at a single hexamer**

The distinct, related, Class 2 structures and their similarity to eukaryotic CMG (Fig. 6) suggest a sequential DNA melting mechanism based on successive conversion of subunit interfaces from loose to tight. The process begins with 6 loose interfaces and all subunits in the horseshoe conformation, represented by the C-tier of Fundamental Class 1. This form is consistent with a nucleotide-free MCM:DNA species that would exist after the necessary exhaust of ADP from the Mcm2–7 double-hexamer[38]. Successive conversion of adjacent interfaces to a tight conformation with associated h2i helix conformations drives sequential binding of one DNA strand by the ps1β and h2i hairpins to sequentially push the aromatic wedge into the DNA minor groove and increase DNA melting. The interfaces are converted from loose to tight by ATP-binding, as shown to occur during eukaryotic CMG activation[38]. Our structures show the disposition of MCM:DNA with 0, 2, and 3 tight interfaces. We have not detected the presence of a species with one tight interface among the cryo-EM particles, but we presume that such a species must exist so that multiple ATP-binding events are not precisely concerted. Although we have not identified a structure with a single tight interface, the comparative RMSD analysis revealed a candidate structure (PDB 9E2X[98]) of CMG encircling one strand of DNA with a G-quadraplex and a 3′-ssDNA extension (Fig. 8a). This structure had one tight interface according to the distance criteria (Table 1) and adopts the anticipated setting for a sequential mechanism (see below). Two of the h2i aromatic side-chains, Mcm2 and 6, stack on nucleotides of the ssDNA, but the aromatic side-chain of Mcm5 sits below the DNA. This structure does not allow assessment of the degree of melting

because only one strand is present. The structure of loading intermediate 9GJW[89] has a low RMSD in the same setting as 9E2X[98] (Supplementary Table 8) and shows a similar degree of engagement of the h2i aromatic residues with dsDNA that is not melted (Fig. 8b). Two of the h2i aromatic side-chains, Mcm2 and 6, are projected to the minor groove, but the aromatic side-chain of Mcm5 is more distant from the DNA.

The first mechanistic step is to tighten one interface of a C-tier with 6 loose interfaces (represented by Class 1a) to a structure that resembles that of PDB 9E2X[98] (Fig. 9a). Tightening a second, adjacent, interface yields the Class 2a species with two tight interfaces and two DNA base-pairs melted (Fig. 9a). This MCM:DNA species is also represented by the 7PMN[96] structure. Conversion of another adjacent interface to tight yields the Class 2b structure with three tight interfaces and 4 bases of DNA melted (Fig. 9a). This structure strongly matches the eukaryotic melted DNA species that follows GINS recruitment (PDB 7Z13[71]) and matches the stable eukaryotic staggered Form II structure (Fig. 4).

For this sequential mechanism, the transition from Class 2a to Class 2b involves tightening one interface that is adjacent to two consecutive tight interfaces. The tightened interface could be the first empty interface clockwise or counterclockwise. Considered at the ATP site level, the ATP site that is tightened could involve the Walker-A site of one of the subunits participating in tight binding or the arginine finger of one of these subunits. These scenarios are not distinguishable for archaeal homo-hexameric structures because the subunits are identical in sequence. The comparative structures of PDB 7PMN[96] and 7PMK[96] definitively show the directional order because they derive from heterohexameric structures with distinguishable subunits. These structures illustrate that the tightened interface is at the counter-clockwise side (3/5 interface tightened rather than 6/4; see Table 1 and Fig. 6). Notably, the directional order of ATP-binding during sequential melting is identical to the directional order of ATP-binding in subsequent DNA translocation[31].

**Double-hexamer consideration**

Single CMG hexamers are sufficient to provide limited DNA unwinding[117] that is consistent with the structural species described above. Extended origin unwinding, which is an essential component of replication initiation, requires formation of a competent double-hexamer[117]. The structure of the human Mcm2–7 double-hexamer encircling dsDNA (PDB 7W1Y[99]) includes a melted DNA bubble at the

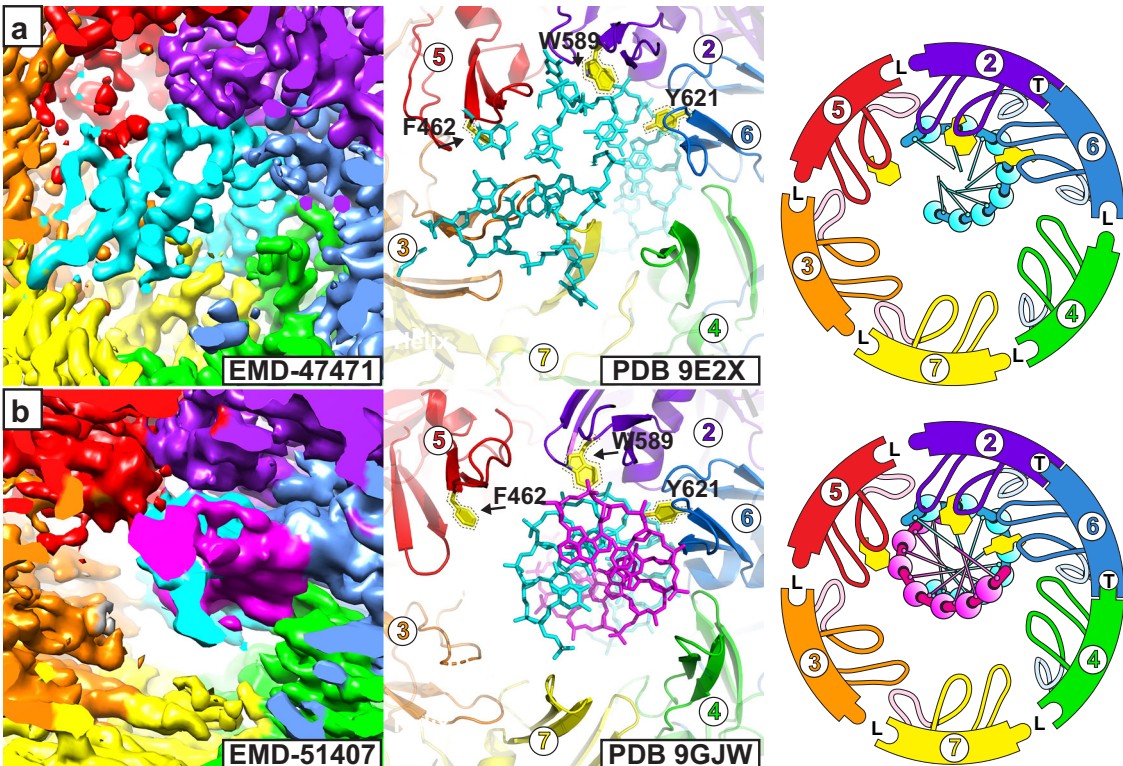

**Fig. 8 | Mcm2–7 structures that illustrate initial, partial engagement of the DNA-melting wedge. a** Yeast CMG bound to G-quadraplex with a 3′-ssDNA tail (PDB 9E2X[98]; EMDB-47471[98]) has the aromatic group of Mcm6 Y621 stacked on a DNA sugar and base as in the structures of Fig. 6. The aromatic group of Mcm2 W589 does not extend over the DNA sugar and base as far as in the structures of Fig. 6, and the aromatic group of Mcm5 F462 does not stack on the DNA. **b** The ATPase tier encircling undistorted dsDNA in PDB 9GJW[89] (EMDB-51407[89]) similarly shows the aromatic group of Mcm6 Y621 stacked on a DNA sugar and base, the aromatic group of Mcm2 W589 does not extend over the DNA sugar and base as significantly, and the aromatic group of Mcm5 F462 is distant from the DNA. Structure chain identifiers are provided for each model in the middle and right panels (Mcm2: purple 2, Mcm6: blue 6; Mcm4: green 4; Mcm7: yellow 7; Mcm3: orange 3; Mcm5: red 5). Right panel cartoons depict the two hairpins as in Figs. 5 and 6. H2i aromatic groups are shown in yellow for right panel cartoons.

center of the complex at the double-hexamer interface. The DNA properties that make it susceptible to forming a bubble may be an important part of origin identification that precedes the DNA melting at the ATPase tiers. The central DNA bubble of the human structure (PDB 7W1Y[99]) could potentially migrate to the C-tier based on activities of the MCM ring, but we feel that the DNA replication initiation mechanism will generally tolerate re-annealing of this central DNA bubble. The central bubble (PDB 7W1Y[99]) seems to at least partly derive from the double-hexamer architecture because the corresponding single-hexamer structures (PDB 8S0A[114] and PDB 8W0E[94]) are very similar in underlying hexamer:DNA arrangement, but they do not exhibit melted DNA at the N-tier. The central N-tier bubble of PDB 7W1Y[99] thus appears to derive from simultaneous interaction of each constituent hexamer C-tier with DNA in conjunction with a double-hexamer interaction that enforces a greater distance between the C-tiers than undistorted B-DNA allows. If any of these three interaction modes are lost, the additional degrees of freedom would likely allow the DNA to re-anneal. The exhaustion of nucleotides from the MCM complex at the onset of DNA replication is expected to release the grip of both hexamers on DNA. As illustrated in the Class 2 and comparative structures above, only MCM subunits that participate in a tight interface appropriately position DNA-binding modules of adjacent subunits for binding DNA. Exhaustion of all nucleotides from the MCM ATPase sites is expected to convert all interfaces to loose with concurrent loss of interactions with DNA (analogous to the Class 1a structure). Although the hexamers would continue to encircle DNA, the lost grip would likely allow the central bubble to re-anneal.

To demonstrate our preferred path for the sequential melting mechanism of Fig. 9a in the context of a double-hexamer, a molecular morph movie was generated using the rigimol routine of PyMOL[86]. The process begins with one hexamer losing its grip on DNA and sliding away from the other hexamer, breaking the double-hexamer interface. The sliding hexamer's lost grip on DNA allows the central DNA bubble to re-anneal as described above. The constituent hexamers are then converted to the melting form (eukaryotic staggered Form II) by the sequential melting mechanism of Fig. 9a. We illustrate the transitions at each hexamer independently to emphasize that the transformations among the ATPase sites are likely not precisely concerted. Following conversion of both hexamers, the final melting structure of PDB 7Z13[71] is established (Supplementary Movie 2). The morph movie is intended to provide general qualitative visualization of our preferred path for sequential melting and not to convey specific molecular dynamics.

**Proposed mechanism for a fixed MCM aromatic wedge to extend origin unwinding**

The specific placement of aromatic wedge residues at Mcm5, 2, and 6 and absent from Mcm4, 7, and 3 profoundly implies that the MCM ring does not extend initial DNA melting via the rotary translocation mechanism[31] that the mature helicase uses at the replication fork to pull one encircled DNA strand through its central pore. For translocation and melting activities, MCM subunits (predominantly) bind one strand of DNA in a spiral staircase (see Fig. 7a, b). In the rotary DNA translocation activity, the collective subunit staircase moves down, pulling the bound DNA through the central pore. Next, the subunit at the bottom of the staircase releases its grip on DNA and moves to the top to bind the next incoming nucleotide of ssDNA. The binding and release events proceed around the ring, with each subunit sequentially

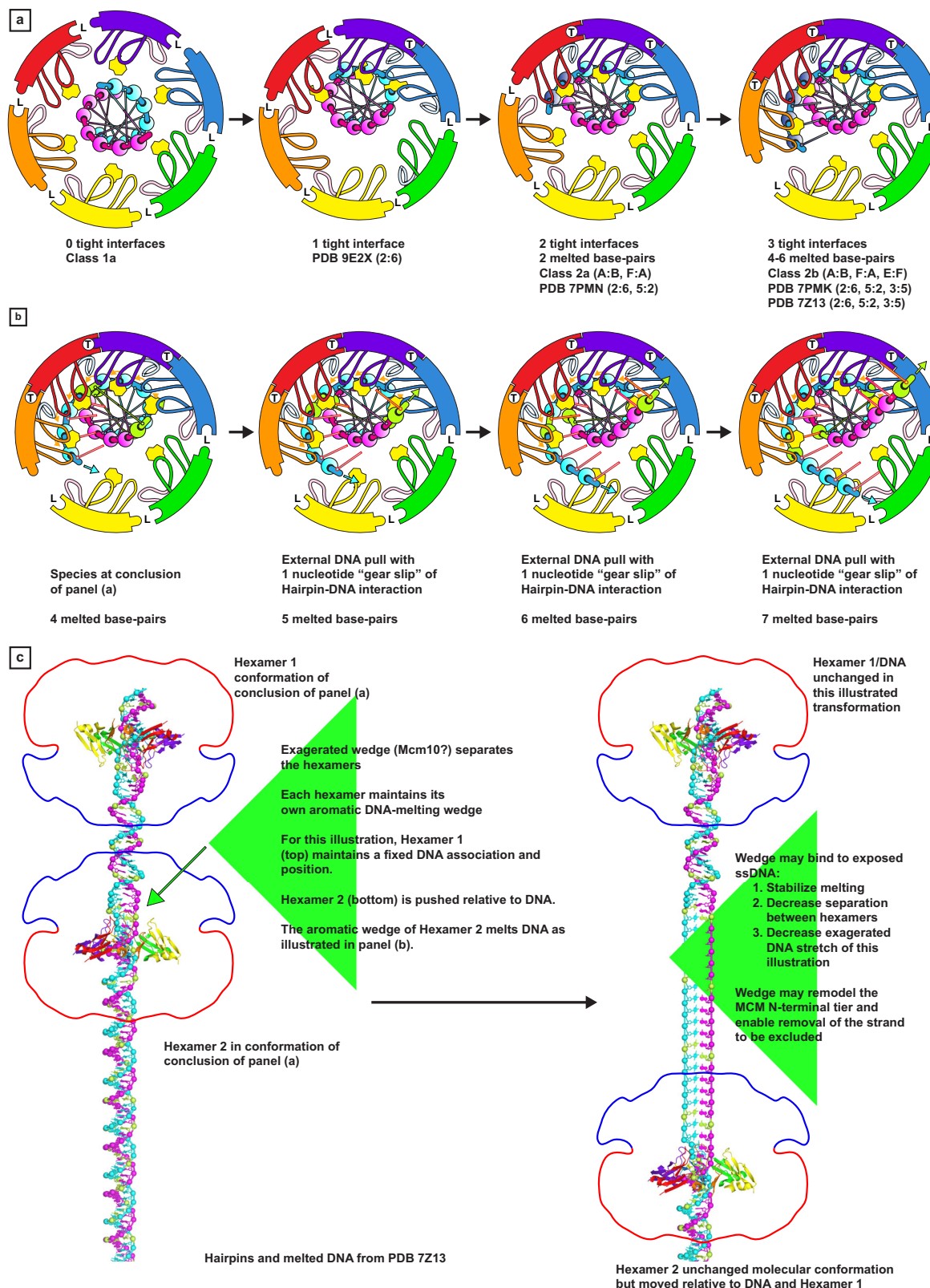

taking its turn at the top of the staircase. This activity is possible for the eukaryotic heterohexamer in CMG because all Mcm2–7 subunits conserve the polar ssDNA-contacting residues of the ps1β and h2i. In contrast, only Mcm5, 2, and 6 have aromatic wedge residues, and hence the other subunits (Mcm4, 7, and 3) cannot take a turn to serve as an aromatic wedge. A sequential rotary action in the melting complex (such as PDB 7Z13[71]) will convert it to a form where aromatic wedge subunits no longer participate in tight interfaces or interact with DNA and thus will not be able to melt DNA.

We therefore suggest that the overall protein complex of the melting conformation (Class 2b, PDB 7PMK[96], PDB 7Z13[71], PDB 7QHS[71]) is static with an aromatic wedge persistently poised to melt DNA. In this form, melting can be extended if an external force pulls the DNA relative to the protein (Fig. 9b and Supplementary Movie 3). An

**Fig. 9 | A mechanism for DNA-melting by sequential conversation of subunit conformations. a** The process begins with all interfaces loose (L) as in the Class1a structure. Sequential tightening of interfaces provides increasing engagement of the DNA-melting wedge (yellow) of three adjacent subunits with increased melting of the DNA. Melted DNA bases are indicated by dark blue shaded spheres. Representative structures and the chain identifiers of associated tight interfaces are noted below each step. The final step corresponds to the Staggered Form II structure that is highly stable based on its abundance in the PDB (see Fig. 4). **b** DNA melting can be extended if an external force pulls the DNA with respect to the stable Staggered Form II structure with its fixed DNA-melting wedge. For a discrete step, the DNA slides one increment along the DNA-binding hairpins analogous to a gear slip (orange arrows, see Supplementary Movie 3). A set of nucleotides are highlighted in green to highlight the transformation from base-paired to melted and DNA movement with respect to the ring. Unpaired bases are outlined in red. **c** An external force that pulls the DNA relative to the static MCM ring could be provided by an external wedge, perhaps Mcm10, that pries the two hexamers apart. This wedge may also bind and stabilize exposed ssDNA and assist to remodel the MCM N-tier to enable subsequent exit of the strand to be excluded from the MCM ring during active replication (see Supplementary Movie 4). A hypothetical wedge is illustrated as a large green triangle. The hairpins and melted DNA locations are derived from the coordinate file of PDB 7Z13[71]. Molecular model images and green triangles were generated by PyMOL[86].

alternative description is that melting can be extended by an external force that pushes the aromatic wedge, like a plow, into undistorted DNA and sequentially melts one base-pair at a time. In this mechanism, the DNA nucleotides all move one sequential position in their arrangement with the MCM DNA-binding hairpins, which contrasts the continuous association of each DNA nucleotide with a specific hairpin that occurs during rotary translocation. If the DNA is viewed as a chain, and the MCM DNA-binding hairpins as teeth of a sprocket, the rotary translocation mechanism follows a standard chain drive where the teeth maintain continuous association with specific chain links, and rotation of the sprocket drives the entire chain to move. In the above mechanism for extension of DNA melting, the sprocket remains fixed while an external force pulls the chain such that it slips over the sprocket in one increment. Based on this analogy, we refer to this as a gear slip (Fig. 9b and Supplementary Movie 3). Notably, this melting extension is not based on ATP hydrolysis by MCM because the MCM ring is static, and the MCM:DNA movement is caused by an external factor.

A likely candidate to provide an external force is Mcm10. First, Mcm10 has been shown to function as a wedge that pries the constituent hexamers of the double hexamer apart[72]. Second, Mcm10 is the decisive factor to transition from initial melting to extensive origin unwinding[38]. With both hexamers in a static melting conformation, as illustrated by PDB 7Z13[71], the action of prying the two hexamers apart from each other will directly push the aromatic wedge into the undistorted DNA beyond the C-tier of the hexamer (Fig. 9c and Supplementary Movie 4). Notably, this movement of the MCM ring with respect to the DNA is opposite in direction to that used during rotary ssDNA translocation[31]. In the absence of contrary information, we presume that archaeal MCM will melt DNA with a mechanism that is highly similar to that of eukaryotes with an external force pushing a static aromatic wedge. Although an archaeal counterpart of Mcm10 has not been identified[118], we predict that an unidentified factor pushes the MCM ring in the static melting conformation towards undistorted DNA (or pulls the DNA over the wedge) to extend DNA melting for replication initiation.

### Lagging strand exit from the ring

We suggest that final exit of the lagging strand from the ring occurs stepwise at the two tiers because all CMG structures adopt a staggered conformation (Fig. 4). The staggered conformation cannot readily develop a contiguous opening across both tiers to enable a strand to exit both tiers in a single step. All staggered conformation structures have a closed ring at the N-tier. In contrast, the staggered conformation is able to generate a wide gap at the C-tier as illustrated by our earlier ssDNA translocation structure where one ATPase domain is disordered (PDB 8EAM[16]). This gap is sufficiently large to enable the lagging strand to pass through and exit the C-tier ring, and the gap is naturally re-sealed by the rotary DNA translocation activity of the ATPase tier[16]. We suggest that such lagging strand exit from the C-tier is preceded by its exit from the N-tier, and such N-tier exit may occur when Mcm10 wedges the two hexamers apart[72] with Mcm10

remodeling the MCM ring[119,120] at the N-tier to enable strand exit. Mcm10 may also bind to exposed ssDNA[121] to assist strand exit.

### Biological context

The single-hexamer structures reported here show progressive melting at the end of DNA. Biological DNA replication begins with an MCM double-hexamer at interior regions of DNA and requires multiple firing factors and additional protein recruitment[38,61,71]. The exclusive MCM:DNA composition of the samples reported here assures that the DNA melting is mediated by the MCM proteins, but this composition does not show DNA at the N-terminal tier for a melted DNA complex, which would be present when melting occurs an interior region of DNA[71]. The structures of PDB 7PMN[96] and PDB 7PMK[96] similarly show CMG with DNA melted at an end. Within these structures, the MCM ring likely sits at the end of the DNA because it is more energetically favorable to melt DNA at the end than in the middle. In contrast, the structure of PDB 7Z13[71] was generated on a piece of DNA flanked at both ends by nucleosomes to enforce an interior DNA positioning, and this structure uniquely shows CMG with DNA melted by the C-tier at an interior DNA region.

The homo-hexameric construction of the archaeal MCM hexamer positions conserved aromatic wedge residues equivalently in all settings. As such the distinct, stable staggered forms of eukaryotic Mcm2–7 (Fig. 4, Staggered Form I and Staggered Form II) may not be possible for archaeal MCM. The additional form in eukaryotes is that of the stable and idle MCM:DNA double-hexamer and suggests that a stable staggered form that does not melt DNA does not exist in archaea. Development of the capacity to generate these two fundamental, stable states is likely part of the basis for the evolution of the MCM helicase from homohexamers in archaea to heterohexamers in eukaryotes.

## Methods

### Cloning, expression, and purification

The previously described protein construct (pJM001.3[29]) is a chimeric fusion of the *Saccharolobus solfataricus* MCM N-terminal domain and the *Pyrococcus furiosus* MCM AAA+ domain (*Sso-Pf*MCM) with an N-terminal His$_6$-SUMO tag. The original SUMO vector was the generous gift of Dr. Christopher D. Lima[122]. The *Pyrococcus furiosus* genetic sequence has an intein in the AAA+ domain (*Pf*MCM amino acids 362–728[48]) that was genetically removed from the construct[29]. The protein construct expresses *Sso*MCM residues 1–269, followed by *Pf*MCM residues 257–361, and then *Pf*MCM residues 729–966[29]. Freshly transformed BL21(DE3)-RIPL chemically competent cells were grown overnight in a 100 mL starter culture of LB media with 30 mg/L kanamycin and 2% glucose. The starter culture was distributed to 6 L of LB media containing 0.4% glucose and 30 mg/L kanamycin. The cultures were grown at 37 °C to O.D. 0.3 and subsequently grown at 18 °C. IPTG was added to 0.4 mM at O.D. 0.7 followed by overnight growth at 18 °C. Cells were harvested by centrifugation, resuspended in 50 mM Tris pH 8.3, 250 mM NaCl, 10% glycerol, 5 mM β-ME, and lysed with a microfluidizer. The soluble fraction was isolated by centrifugation. Nucleic

acids were removed by adding 5 M NaCl to 1 M final concentration followed by 10% polyethylenimine-HCl to 0.3% final concentration. The soluble fraction was isolated by centrifugation, and the protein was precipitated by adding 43.6 g of ammonium sulfate per 100 mL. The precipitated protein was isolated by centrifugation and resuspended in 50 mM Tris pH 8.3, 500 mM NaCl, 10% glycerol, 10 mM imidazole, 5 mM β-ME followed by batchwise binding to Ni-NTA agarose (Qiagen) with five subsequent washes with the same buffer. The protein was eluted with 50 mM Tris pH 8.3, 500 mM NaCl, 10% glycerol, 250 mM imidazole, 5 mM β-ME, and subjected to size-exclusion chromatography in 20 mM HEPES pH 7.6, 200 mM NaCl, 5 mM β-ME, eluting at a volume consistent with a hexamer. Ulp1 protease was added, and the sample was dialyzed overnight against 20 mM Tris pH 8.3, 100 mM NaCl, 5 mM β-ME to cleave the SUMO tag. The Ulp1 protease plasmid was the generous gift of Dr. Christopher D. Lima[122]. The cleaved MCM protein was purified by anion exchange with a 0.1–1 M NaCl linear gradient in 20 mM Tris pH 8.3, 5 mM β-ME. Selected fractions were purified by gel-filtration chromatography in 20 mM HEPES pH 7.6, 200 mM NaCl, 5 mM β-ME; dialyzed against 25 mM HEPES pH 7.6, 10 mM NaCl, 5 mM magnesium acetate; and concentrated to 20 mg/mL for storage.

### Cryo-EM sample preparation, data collection, and processing

Cryo-EM samples were prepared to consist of 2 mg/mL Sso-PfMCM, 5 mM ATP-γS, 0.25 mM 60-mer palindromic DNA substrate (see below for specific sequences) in 25 mM HEPES pH 7.6, 10 mM NaCl, 5 mM magnesium acetate. Samples were incubated for 2 h at room temperature and then passed through a 0.22 μm spin filter. Filtered samples were plunge frozen on a Vitrobot using C-flat grids. Cryo-EM data were collected using a Titan Krios (Thermo Fisher) transmission electron microscope, equipped with a K3 direct electron detector and post column GIF (energy filter). K3 dark and gain reference were collected just before data collection. Data collection was performed in SerialEM software[123] with image shift protocol (9 images were collected with one defocus measurement). Movies were recorded at defocus values from −0.8 to −1.8 μm at a magnification of 81kx, which corresponds to the pixel size of 1.08 Å at the specimen level (super resolution pixel size is 0.54). Motion correction was performed on raw super resolution movies stacks and binned by 2 using MotionCor2 software[124]. For each structure, an initial evaluation of a subset of the micrographs was performed with Relion[125], CTFFind[126], cisTEM[127], Cryosparc[128], pyem[129], and Chimera[130].

For the Sso-PfMCM:Mg:ATP-γS:DNA1 complex, the protein solution and was combined with a 60-mer palindromic oligonucleotide that contained a repetitive sequence, ACTGACTGACTGACTGACT-GACTGACTGACGTCAGTCAGTCAGTCAGTCAGTCAGTCAGT. This DNA sequence was inspired by the deposited oligonucleotide sequence of the cryo-EM structure of eukaryotic Mcm2–7 double-hexamer encircling duplex DNA (PDB 5BK4[37]). The sample was frozen to grids with a vitrobot, and 5,668 micrographs were collected with the following parameters: Magnification: 81.000; Pixel size: 1.08 (Counting)/0.54 (Super Resolution); Exposure time: 3 s/0.05 s per frame/60 frames; Dose: 29.74 e⁻/physical pixel/second or 25.5 e⁻/Å²/s; Total dose: 76.5 e⁻.

The motion-corrected micrographs were imported to Cryosparc[128], and its Patch CTF routine was applied to each micrograph. 45,565 particles were picked by the Cryosparc[128] Blob picker routine from 50 micrographs. Particle images were extracted and used in 2D classification to generate templates that were used in the Cryosparc[128] Template Picker for the full dataset. 15,146,308 particles were identified, and their positions were converted for use in Relion[125] by pyem. In Relion, the micrographs were imported, and CTF parameters were determined using CTFFind[126]. The imported particle positions were used to extract particle images at 8.64 Å per pixel. These particle images were imported to Cryosparc[128], subjected to 2D

Classification, and 4,851,305 particles were selected. These particle positions were converted for use in Relion[125] by pyem[129] and were extracted at 4.32 Å per pixel by Relion. The particle images were imported to Cryosparc[128], subjected to 2D Classification, and 3,369,871 particles were selected. These were extracted at full 1.08 Å per pixel resolution by Cryosparc[128] to yield 3,304,031 particles. These particles were subjected to two rounds of 2D classification followed by duplicate removal in Cryosparc[128] to yield 2,388,484 particles for use in subsequent refinement. These particles were subjected to Heterogeneous Refinement in Cryosparc[128] with 6 classes. Two of the classes had hexameric ring structures with obvious DNA in the central channel: Class 1 (DNA passing through both tiers, 353,074 particles) and Class 2 (DNA at C-tier, 579,241 particles).

Class 1 was subjected to homogeneous refinement followed by 3D-classification with 10 classes in Cryosparc[128]. Ten volumes were used to seed 10 classes of Heterogeneous Refinement in Cryosparc[128]. One of these subclasses (74,910 particles) had six well-ordered ATPase domains and appeared C6-symmetric with cylindrical, featureless DNA. This subclass ultimately generated the Class1a structure by symmetry expansion and 3D classification (40,953 particles; described further in section below on Merged particles). The final refinement was based on Local Refinement in Cryosparc[128]. Two additional subclasses had only five well-ordered ATPase domains and thus intrinsically asymmetric. An initial homogeneous refinement in Cryosparc[128] (95,316 particles) showed DNA strand features, but the DNA structure was noisy and suggested multiple conformations. These refined particles were subjected to 3D classification in Cryosparc[128] in 6 classes using a cylindrical focus mask that encompassed the central channel. This classification revealed conceptually similar structures where the h2i of a subunit of an ATPase domain located adjacent to the disordered ATPase domain significantly approached one of the DNA strands, but the polarity of this interaction strand differs. These two subclasses were refined and partitioned as described under Merged particles. The refined particles of Merged particle refinement were separated by dataset for final set-specific refinements. Class1b was based on Cryosparc[128] Local Refinement of 32,139 particles, and Class1c was based on Cryosparc[128] Local Refinement of 15,877 particles.

Class 2 was subjected to homogeneous refinement followed by 3D-classification with 10 classes in Cryosparc[128]. The particles from 9 of 10 classes (517,028 particles) were pooled and used in Heterogeneous Refinement in Cryosparc[128] seeded with 4 of the volumes from the prior 3D-classification job. This refinement distinguished two forms within the C-tier, Class2a with 4 horseshoe helix-2-inserts (148,429 particles) and Class2b with 3 horseshoe (196,176 particles). The final refinements for each were obtained by Homogeneous Refinement in Cryosparc[128].

Class 3 (heptamer) was obtained by Cryosparc[128] Local Refinement (25,549 particles) using the structure derived from the Merged Particles heptamer (see below) after separating the particles by dataset.

The Sso-PfMCM:Mg:ATP-γS:DNA2 sample was prepared by combining the protein solution with a 60-mer oligonucleotide that contained a random palindromic sequence, TCCGTCTCACAGTAAC AACCCTCCGCCTGATCAGGCGGAGGGTTGTTACTGTGAGACGGA. The sample was frozen to grids with a vitrobot, and 6633 micrographs were collected with the following parameters: Magnification: 81.000; Pixel size: 1.08 (Counting)/0.54 (Super Resolution); Exposure time: 4.9 s/ 0.07 s per frame/70 frames; Dose: 18.829 e⁻/physical pixel/second or 15.963 e⁻/Å²/s; Total dose: 78.22 e⁻.

The motion-corrected micrographs were imported to Cryosparc[125], and its Patch CTF routine was applied to each micrograph. 79,822 particles were identified by Blob Picker from 40 micrographs and extracted. These were subjected to two rounds of 2D classification to generate templates for Cryosparc's Template Picker[128], which identified 14,853,988 particles. These particles were exported to Relion by pyem[129] and extracted in Relion[125] at 4.32 Å per pixel. The

particles were subjected to 3 rounds of Cryosparc[128] 2D classification to yield 4,211,984 particles. The particles were then extracted at 1.08 Å per pixel in Cryosparc[128] to yield 4,174,824 particles, followed by duplicate removal to retain 4,104,256 particles. Following a final 2D classification, 3,793,296 particles remained in the set.

These particles were subjected to Heterogeneous Refinement in Cryosparc[128] with 6 classes. Two of these classes had hexameric ring structures with obvious DNA in the central channel: Class 1 (DNA passing through both tiers, 527,777 particles) and Class 2 (DNA at C-tier, 879,551 particles).

Class 1 was subjected to homogeneous refinement followed by 3D-classification with 10 classes in Cryosparc[128]. The 10 volumes were used to seed 10 classes of Heterogeneous Refinement in Cryosparc[128]. One of these subclasses (110,906 particles) had six well-ordered ATPase domains and appeared C6-symmetric with cylindrical, featureless DNA. This subclass ultimately generated the Class1a structure by symmetry expansion and 3D classification (60,755 particles; described further in section below on Merged particles). The final refinement was based on Local Refinement in Cryosparc[128]. Two additional subclasses had only five well-ordered ATPase domains and thus intrinsically asymmetric. An initial homogeneous refinement in Cryosparc[128] (85,099 particles) showed DNA strand features, but the DNA structure was noisy and suggested multiple conformations. The refined particles were subjected to 3D classification in Cryosparc[128] in 6 classes using a cylindrical focus mask that encompassed the MCM central channel. This classification revealed conceptually similar structures where the h2i of a subunit of an ATPase domain located adjacent to the disordered ATPase domain significantly interacts with one of the DNA strands, but the polarity of this interaction strand differs. These two subclasses were refined and partitioned as described under Merged particles. The refined particles of Merged particle refinement were separated by dataset for final set-specific refinements. Class1b was based on Cryosparc[128] Local Refinement of 28,511 particles, and Class1c was based on Cryosparc[128] Local Refinement of 14,925 particles.

Class 2 was subjected to Heterogeneous Refinement in Cryosparc[128] with 3 classes seeded with volumes corresponding to class2a, class2b, and DNA-free models (see below for seed volume generation description). One class was selected (291,379 particles), subjected to Homogeneous Refinement, followed by 3D classification to generate 10 seed volumes that were used in Heterogeneous Refinement of the 10 classes. Seven classes were selected, pooled, and subjected to a final Homogeneous Refinement (Class 2a, 235,648 particles). A second class from the initial Class 2 Heterogeneous Refinement job was selected (328,400 particles), subjected to Homogeneous Refinement followed by 3D classification to generate 10 seed volumes that were used in Heterogeneous Refinement of the 10 classes. Six classes were selected, pooled, and subjected to a final Homogeneous Refinement (Class 2b, 275,045 particles).

The seed volumes for the first Heterogeneous Refinement of Class 2 were obtained by first subjecting the 879,551 particles of Class 2 to 3D classification in 10 classes. One of these classes appeared DNA-free and formed one seed volume after Homogeneous Refinement. Eight of the classes had obvious DNA at the central channel of the ATPase domain, and these were pooled and subjected to Heterogeneous Refinement seeded by four classes of the prior 3D Classification job. One of these was used as a seed volume following its Homogeneous Refinement (Class 2a), and two appeared similar and were pooled and subjected to Homogeneous refinement, generating the third seed volume (Class 2b).

Class 3 (heptamer) was obtained by Local Refinement (25,290 particles) using the structure derived from the Merged Particles heptamer (see below) after separating the particles by dataset.

## Merged particle structures

For the two datasets, the particles comprising Class 2a and Class 2b were pooled respectively and subjected to Homogeneous Refinement in Cryosparc[128] to generate refined structures based on the merged particles. The other two classes of structure were very nearly symmetric for the protein component and required special classification treatment to elucidate the features of the central DNA that was enhanced by maximizing the total particles. As a result, these computations were performed on the merged particles of the two datasets, and final structures for the individual datasets were then obtained by separating the particles based on the respective datasets.

Class1a is nearly C6-symmetric at the protein level, and classification jobs tended to be dominated by versions of the complex that illustrated varying disorder of the peripheral helical bundle subdomains of the NTD. The following procedure provided classification based on the central DNA features. First, the Class1a particles of the two datasets (185,816) were pooled and subjected to Homogeneous Refinement in C6 symmetry. The resulting particles were sixfold expanded about the C6-axis (1,114,896 particles) and subjected to 3D Classification in 10 classes using a central channel focus mask. The focus mask was created with 5-pixel soft-padding from a cylinder shape (9 Å radius, 100 Å height) created in Chimera[130]. The best-looking class was selected and subjected to Local Refinement (113,110 particles) using the full solvent mask of the previous classification job. Duplicate particles were removed (101,708 particles retained), and the final Class 1a structure was obtained by final Cryosparc[128] Local Refinement. The particles were separated into their respective datasets and generated final structures by Local Refinement (see above).

The particles of the structures of 5 well-ordered ATPase domains for both datasets were combined and subjected to 3D classification in Cryosparc[128] using the cylindrical focus mask (above). Two classes were pooled to generate Class1b (60,650 particles) following Cryosparc[128] Local Refinement. One class of 30,802 particles generated Class1c following Cryosparc[128] Local Refinement. The particles were separated into their respective datasets and generated final structures by Local Refinement (see above).

Class 3 is nearly C7-symmetric at the protein level, analogous to the near C6-symmetry of Class 1a. The smaller number of particles of the heptameric form greatly benefited by merging the two dataset particles in a procedure similar to that of Class 1a. 2D classification indicated a minor presence of heptameric rings in views parallel to the ring channel. All other views were not obviously in a specific oligomeric form, and thus the particles comprising these class averages might contain hexamer, heptamer, or other forms. An initial map for the heptamer was obtained from a subset of particles in Cryosparc[128] by calculating 6 ab initio classes and selecting a class that was recognizable when looking down the channel axis at one side with decreasing order when moving away from that view. This structural class appeared to derive from superimposed particles of hexamer and heptamer. The particles were subjected to homogeneous refinement followed by 3D classification in 10 classes, which improved the density across the ring. The most well-defined class that appeared to encircle DNA was selected and subjected to homogeneous refinement followed by 3D classification in 3 classes. The best class was selected and subjected to homogeneous refinement in C1-symmetry to generate a reference map that was used in a subsequent refinement in C7-symmetry, which aligned the structure to the symmetry axis. The particles were then symmetry expanded about the C7-axis and then subjected to 3D classification in 7 classes using the same cylindrical focus mask used for Class 1a above. The best-looking structure was selected and subjected to Local Refinement using the full solvent mask of the preceding 3D classification job. This heptamer volume provided one class for a large-scale classification job of all the particles of the two datasets.

The full set of initially selected particles for the two datasets were used for Cryosparc[128] ab initio reconstructions in 6 classes. The six generated volumes and the heptameric volume (see above) were used in a 7-class Heterogeneous Refinement job with the full set of particles. The heptameric class (379,160 particles) was subjected to

homogeneous refinement followed by 3D classification in 10 classes. The 10 volume classes were used as the initial volumes in a Cryosparc[128] Heterogeneous Refinement job. The heptameric class was selected (89,337 particles), and the particles were used to generate 3 ab initio classes in Cryosparc[128], and the resulting volumes were used as initial volumes in a 3-class Cryosparc[128] Heterogeneous Refinement. The best class (71,238 particles) was selected and subjected to homogeneous refinement in C1 symmetry, followed by refinement in C7 symmetry. The particles were expanded about the C7-axis and subjected to 3D classification in 7 classes using a cylindrical focus mask (20 Å radius, 100 Å height). The best class was selected and subjected to Local Refinement (61,530 particles) using the full solvent mask of the previous 3D classification. Duplicate particles were then removed (50,889 particles retained), and an additional Local Refinement job was performed. The Cryosparc[128] Particle Sets tool was then used to generate the final particle set (50,839 particles) by removing the particles contributing to Classes 1 and 2. A final Cryosparc[128] Local Refinement job with this particle set generated the structure of heptamer for the merged particle set. The structures for the individual datasets were then obtained by Local Refinement using the particles of the respective datasets.

### Model-building and refinement

Initial models were constructed from the crystal structure of *Sso-Pf*MCM, PDB 4R7Y[29], by placing its N- and C-terminal domains in the respective density. The structures were refined with phenix.real_space_refine[131,132] and rebuilt in Coot[133–135]. The final refined coordinate files contain either ATP-γS or ADP at each site according to the strength of the γ-phosphate in the EM map. The inclusion of ADP molecules in the model is to model the apparent density and does not indicate that hydrolysis of the γS-phosphate has occurred. The refinements included bond distance and octahedral angle restraints for the associated magnesium ion. For ATP-γS, the magnesium was bonded to a Walker-A S335 side-chain, an ATP-γS β-phosphate oxygen, an ATP-γS γ-phosphate oxygen, and three water molecules. For ADP, the magnesium was bonded to a Walker-A S335 side-chain, an ADP β-phosphate oxygen, and four water molecules. The polypeptide construct consists of *Sso*MCM residues 1–269, followed by *Pf*MCM residues 257–361, and then *Pf*MCM residues 729–966[29]. In the deposited molecular coordinate files, these residue spans are numbered 1–269, 1257–1361, and 1729–1966.

### Figure preparation

Figures were prepared with the programs PyMol[86], Chimera[130], ChimeraX[136], and Adobe Illustrator. Residues are numbered with native amino acid numbers without intein removal. Cartoon representations depict MCM hexamers as six-color rings with tight and loose interfaces and DNA as beads on a string, similar to the style we have used previously[16,31].

### RMSD comparisons

A report from the PDB with all structures meeting the search term "mcm" and a 200 kDa molecular weight cutoff was downloaded. For each, a hexameric MCM structure consisting of six 152-amino acid ATPase core[16] structures was constructed by automated alignment in PyMOL[86] of the 152-amino acid ATPase core of chain Class 2b chain F to each chain 2–7 (residues 319:361/729:837 renumbered to a contiguous span, 332–483). An ATPase core hexamer for a candidate alignment model (such as Class 2b) was similarly constructed by automated alignment in PyMOL[86] of the same 152-amino acid ATPase core of Class 2b chain F to each chain A-F, and six different settings were constructed by modifying the chain identifiers in all six permutations of the directional order of eukaryotic subunits around the ring (such as 2, 6, 4, 7, 3, 5—see Supplementary Fig. 12 for illustration of the process including the six settings of Class 2b for candidate alignments). The

RMSD of each setting against each eukaryotic Mcm2–7 structure of the downloaded list was calculated and sorted to compare the collective disposition of the six MCM ATPase cores as a unit. Altogether, 77 structures were compared in 6 settings for a total of 462 comparisons. The top 30 RMSD alignments for Class 2b (Supplementary Table 8) were visually inspected to identify the comparative structures used. The RMSD analysis of Class 1a (Supplementary Table 7) did not identify strong candidates for comparison.

### Dihedral analysis

For each of the 77 structures identified above, an MCM N-tier model was generated by automated alignment in PyMOL[86] of the Class 2b chain F OB-fold (residues 105:130 / 185:265) to each subunit of the candidate Mcm2–7 structure. An MCM C-tier model was generated by automated alignment in PyMOL[86] of the Class 2b chain F 152-amino acid AAA+ core[16] (residues 319:361/729:837 renumbered to a contiguous span, 332–483) to each subunit of the candidate Mcm2–7 structure. The chains for each were renamed to correspond to the Mcm2–7 structure, and centroid positions for each tier were calculated by PyMOL[86]. For each chain of the constructed alignments, a dihedral was calculated by PyMOL[86] from the following four sequential positions (with original Class 2b residue numbering): *Sso*MCM residue P201 C-alpha, OB-fold centroid, AAA+ centroid, followed by *Pf*MCM residue Y755 C-alpha. Five structures did not contain all 6 OB-folds and were excluded from the analysis.

### Reporting summary

Further information on research design is available in the Nature Portfolio Reporting Summary linked to this article.

### Data availability

The molecular coordinates and maps have been deposited at the PDB and EMDB with the following accession codes. 9NUH (SsoPfMCM:DNA class 1a from merged particles). 9NUI (SsoPfMCM:DNA class 1b from merged particles). 9NUJ (SsoPfMCM:DNA class 1c from merged particles). 9NUK (SsoPfMCM:DNA class 2a from merged particles). 9NUL (SsoPfMCM:DNA class 2b from merged particles). 9NUM (SsoPfMCM:DNA class 3 from merged particles). 9NUN (SsoPfMCM:DNA class 1a from DNA 1). 9NUO (SsoPfMCM:DNA class 1b from DNA 1). 9NUP (SsoPfMCM:DNA class 1c from DNA 1). 9NUQ (SsoPfMCM:DNA class 2a from DNA 1). 9NUR (SsoPfMCM:DNA class 2b from DNA 1). 9NUS (SsoPfMCM:DNA class 3 from DNA 1). 9NUT (SsoPfMCM:DNA class 1a from DNA 2). 9NUU (SsoPfMCM:DNA class 1b from DNA 2). 9NUV (SsoPfMCM:DNA class 1c from DNA 2). 9NUW (SsoPfMCM:DNA class 2a from DNA 2). 9NUX (SsoPfMCM:DNA class 2b from DNA 2). 9NUY (SsoPfMCM:DNA class 3 from DNA 2). EMD-49806 (SsoPfMCM:DNA class 1a from merged particles). EMD-49807 (SsoPfMCM:DNA class 1b from merged particles). EMD-49808 (SsoPfMCM:DNA class 1c from merged particles). EMD-49809 (SsoPfMCM:DNA class 2a from merged particles). EMD-49810 (SsoPfMCM:DNA class 2b from merged particles). EMD-49811 (SsoPfMCM:DNA class 3 from merged particles). EMD-49812 (SsoPfMCM:DNA class 1a from DNA 1). EMD-49813 (SsoPfMCM:DNA class 1b from DNA 1). EMD-49814 (SsoPfMCM:DNA class 1c from DNA 1). EMD-49815 (SsoPfMCM:DNA class 2a from DNA 1). EMD-49816 (SsoPfMCM:DNA class 2b from DNA 1). EMD-49817 (SsoPfMCM:DNA class 3 from DNA 1). EMD-49818 (SsoPfMCM:DNA class 1a from DNA 2). EMD-49819 (SsoPfMCM:DNA class 1b from DNA 2). EMD-49820 (SsoPfMCM:DNA class 1c from DNA 2). EMD-49821 (SsoPfMCM:DNA class 2a from DNA 2). EMD-49822 (SsoPfMCM:DNA class 2b from DNA 2). EMD-49823 (SsoPfMCM:DNA class 3 from DNA 2)

### Code availability

Elementary bash scripts to execute PyMOL were used to calculate the dihedral angles and RMSD values of Fig. 4 and Supplementary

Tables 7 and 8. The scripting routines and documentation are available at. [https://doi.org/10.5281/zenodo.18487583] (MCM Setting RMSD Calculations). [https://doi.org/10.5281/zenodo.18487880] (MCM Inter-Tier Dihedral Calculations).

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

## Acknowledgements

We thank Alicia Byrd for helpful discussion. Data were collected in the Cryo-EM center of the Structural Biology Department of St. Jude Children's Research Hospital. This work was supported in part by ALSAC, by R35GM136313 (to E.J.E.) from NIGMS, and by the Winthrop P. Rockefeller Cancer Institute. This research used computational resources housed in the Center for Molecular Interactions in Cancer structural biology core and funded by grants P20GM103429 (PI L. E. Cornett) and P20GM152281 (PI R. L. Eoff) from NIGMS. The content is solely the responsibility of the authors and does not necessarily represent the official views of the National Institutes of Health.

## Author contributions

S.R., A.M., and E.J.E. designed and performed the study. Data collection and initial analysis were performed at St. Jude. Structure refinement, analysis, and manuscript preparation were performed at UAMS. All authors contributed to writing the manuscript.

## Competing interests

The authors declare no competing interests.
