## [Transparent Peer Review file · Nature Communications]

Archaeal and eukaryotic MCM rings sequentially melt DNA for replication initiation

Corresponding Author: Dr Eric Enemark

Version 0:

Reviewer comments:

Reviewer #1

(Remarks to the Author)

Rasouli, Myasnikov and Enemark report a sheer number (18) structures of a duplex DNA bound protein chimera with a fusion of the Sulfolobus MCM N-terminal oligomerisation domain and the Pyrococcus MCM ATPase motor domain. Enemark has shown previously that this protein is catalytically active. Its use in structural biology studies already provided insights that proved precious for the field over time. The current work constitutes an important addition to this body of work. Here the authors report various mode of duplex DNA engagement for a single MCM ring. While the authors propose a mechanism whereby the structural transitions in this single hexamer could lead to DNA melting, they choose not to acknowledge some recent work on the mechanism of loading of a single MCM ring, and the structures of a fully loaded single hexamer around duplex DNA, from both human and yeast MCM (though, admittedly, the latter is only a bioRxiv uploaded manuscript). The recent identification of a single duplex DNA loaded MCM in humans was a big deal (with two high profile papers published simultaneously). The discovery that this is a stable intermediate also in archaea is remarkable and makes these data worth publishing in Nature Communications. That said, a rework of the introduction and discussion is recommended, which will provide a proper framework for the discoveries reported. Specific points are detailed below.

Major issues

1. Is it correct to define Class 2 as a DNA melting intermediate, considering the discovery from Steve Bell's lab (Champasa et al 2019) that a yeast MCM variant that cannot homodimerise can form CMG but not open DNA? Would it not be best to compare this structure to the human MCM single hexamer (Weissmann et al 2024 and Yang et al 2024)?
2. The authors observe Class 2 where the leading but not the lagging strand template is visible. They interpret this as a melting intermediate. This could also reflect the fact that the leading strand is visible because it is stabilised by pore loop contacts. This is indeed the case for a helicase loading intermediate (single loaded yeast MCM hexamer) described in a bioRxiv manuscript by Miller, Diffley and colleagues. The PDB coordinates (8RIG) can be downloaded from the PDB already. This is an intermediate on the path to double hexamer formation. In the final product (the DNA is not melted).
3. Along the same lines, I feel the authors should do a better job at drawing a parallel between the mechanism of MCM loading onto duplex DNA and the mechanism of DNA melting by the CMG. In this respect, I note that no reference was made to recent work published in Nature Communications by Butryn et al (2025), which describes a single-loaded yeast hexamer in different duplex DNA binding states, captured by using inactivating ATPase mutants. These structures show a DNA engagement state similar to that of the DNA-melting (or single-stranded DNA translocating) CMG, concluding that the mechanism of MCM interaction and translocation along DNA is always the same, and implying that melting occurs when two CMGs are facing one another. I feel this is another example of a missed opportunity to show how the structure of an archaeal single loaded hexamer on duplex DNA is relevant to mechanistic studies in eukaryotes (both for DNA melting, as stated in this manuscript, but also for helicase loading).

Smaller issues

1. The authors claim that no contacts exist between MCM and DNA in the class 1 structure. They should clarify how the DNA register is established such that duplex DNA can be averaged in the structures. If not contacted by the ATPase domain, duplex DNA must surely be engaged by elements of the N-terminal oligomerisation domain. Otherwise, DNA would be lost from the central channel or only result in a featureless sausage through single-particle averaging. You could refer to studies on the FANCI-FANCD2 DNA encircling ring by Alcon et al Nature 2024, Fig. 4a and Fig. 4b for comparison.

2. Heptameric MCMs had been previously observed in studies by Jose' Maria Carazo, Silvia Onesti and Laura Spagnolo. Their work should be cited.
3. The resolution of the structures reported is high enough for discriminating nucleotides in the ATPase pockets but the authors showed not to show them. I think this should be rectified.
4. Line 37 page 2 "Similar, but different subunits" sounds odd. To me, similar implies that they are different, otherwise they would be identical.
5. Figure 1 is very similar to Extended Data Figure 2 and 3. Figure 2 appears very similar to Extended Data Figure 4 and 5. Could the authors check whether any panels have been mistakenly misplaced?
6. Page 3 line 72 "GINS recruitment provides 1.5–2 turns of DNA unwinding at each CMG" is incorrect. Douglas 2018, which the authors cite in this passage reads "each MCM hexamer untwists around 0.6–0.7 turns of DNA".

Reviewer #2

(Remarks to the Author)

A large number of biochemical and cryo-EM studies in the past few years have focussed on the molecular characterisation of the initiation stages of DNA replication. Progress in this area has been significantly advanced by the ability to reconstitute the process *in vitro* using purified yeast proteins, and solve structures of various intermediate states. Although much is now known about this process, a complete molecular explanation of the very first stages of DNA melting is still lacking. This manuscript focusses on this key question, using the archaeal homo-hexameric MCM replicative helicase as a model system.

The authors present multiple cryo-EM structures of an archaeal MCM single hexamer bound around DNA. Their sample preparation method facilitates the capture of multiple different forms of the MCM-DNA complex, either containing B-form double stranded DNA, or in various states that the authors propose mimic the first stages of replication origin melting. By integrating these structures with previously published structures of archaeal and yeast MCM complexes, the authors propose a compelling model for DNA melting, involving sequential conversion of inter-subunit interfaces from loose to tight states. The manuscript is well structured and written, and clearly explains some quite complex structural conclusions and hypotheses. Further interrogation of the structural data using functional experiments (e.g. with specific mutants *in vitro* or *in vivo*) is completely lacking. Overall, however, given the high quality of the structural data and the general interest in this area of the replication field, I would suggest that the structures alone are sufficiently interesting to warrant publication without functional validation, particularly given the difficulty of such experiments in the archaeal system.

Major points:

- The author's models are heavily dependent on nucleotide-dependent transitions between loose and tight states at inter-subunit interfaces in the MCM ring. To this reviewer, it was unclear whether the authors actually had high enough resolution in their structures to model nucleotide in any of the MCM active sites? If not, this should at least be stated.
- The model in Fig. 5 is dependent on sequential transitions in inter-subunit interfaces around the MCM ring. One tight interface appears to favour the conversion of the neighbouring interface from loose to tight. Can the authors explain how this would work? Is there a structural explanation for why ATP binding at one interface would, for example, favour ATP binding at the adjacent interface?
- Recent structures have demonstrated that a small section of DNA, positioned at the inter-hexamer interface, is already distorted in the human MCM double hexamer (e.g. PDB:7W1Y). Thus, unlike in yeast, melting (or at least distortion of the duplex DNA) has already been initiated before CMG assembly. On the one hand, the authors make almost no mention of the human structures. The authors repeatedly refer to 'eukaryotic' structures, when in reality they are only referring to yeast. Given that a major selling point of this manuscript is the derivation of a generalised model of MCM-driven melting, the authors need to properly consider the implications of the human MCM structures for their models. For example, how do the authors envisage that the initial bubble of distorted DNA trapped near the N-tier of MCM (in the human structures) is converted to melted DNA in the MCM C-tier of each single hexamer, as has been demonstrated previously by the Diffey-Costa groups, as well as in this manuscript?
- In the yeast system, the ejection of ADP from MCM and the initiation of origin melting is dependent on the conversion of MCM double hexamers into pairs of CMG helicases, driven by multiple so-called firing factors (Douglas et al., Nature, 2018). Given this, it is quite surprising that the authors can observe DNA melting by simply loading an MCM single hexamer onto double-stranded DNA in ATP S. Do the authors think this reflects a biologically meaningful difference between the ways in which melting is initiated in archaea and eukaryotes?
- The authors prepare their samples in the slowly hydrolysable ATP analogue ATP S. Presumably the origin melting they observe should be dependent on ATP hydrolysis? What do the authors observe if they prepare their complexes in non-hydrolysable nucleotide (e.g. AMP-PNP)?
- Although the authors are primarily using the archaeal system as a model to understand general principles of MCM function, the lack of discussion regarding the biological mechanisms of DNA replication initiation in archaea seemed an oversight. The authors should describe what is known about the various steps in MCM loading and activation (e.g. as part of the introduction). This is also relevant to point 4 (above).

Minor points:

- Lines 312-315: the authors note that the archaeal structure with 4 tight interfaces also a disordered subunit, potentially providing an exit route for strand exclusion. Is the same true for eukaryotic CMGE (PDB: 7QHS)? If not, does this reflect mechanistic differences between the archaeal and eukaryotic systems?
- Why do the authors use a chimeric MCM fusion, with different domains from different archaeal species?

- More detail should be given on the conditions (concentrations, buffers, times, temperatures) used for preparing the samples before grid preparation
- The different coloured circles that the authors use to explain the different subunit conformations was barely visible on my printed version

Reviewer #3

(Remarks to the Author)

Rasouli et al have investigated the DNA unwinding ability of a hybrid archaeal MCM helicase. The authors employed double-stranded DNA for their cryo-EM analysis of the protein-DNA complexes. ATPγS was employed to slow down ATP-hydrolysis. The sample yielded multiple complex variants. Two classes were observed – one, where dsDNA is observed in the NTD and CTD, the MCM CTD and NTD are twisted, and ATO occupancy is low. The second class, with partially unwound DNA in the CTD, the MCM CTD and NTD are not twisted and nucleotide is observed in several subunits. Within each class, several variants are observed. Class 1 variants detail CTD flexibility, while class 2 variants have different numbers of nucleotides bound and unwound nucleotides. The work is exciting, as it shows the stepwise activation of an MCM hexamer. One could complain that the MCM complex is a hybrid between two different species, but I would argue that the construct is well suited to decipher a mechanism. Nevertheless, some justification should be given. The structure is well described, but figures must be improved to highlight key structural changes, especially for a general audience. Also, the processing workflow is missing and the identified changes could benefit from some verification. The comparison to eukaryotic MCM is well done.

Major comments:

Could the authors test their DNA unwinding model biochemically e.g. by mutating S357 or similar.

The authors need to provide a detailed EM data processing workflow.

The description of the main structural changes e.g. eclipsed vs staggered need to be illustrated by a cartoon. The vertically oriented α-helices are mentioned but not clearly shown in any figure. The structural changes need to be conceptualised and shown in the form of a cartoon.

Minor comments:

In the context of the staggered and eclipsed conformation, the eukaryotic should be discussed more, here, MCM2-7 NTD/CTD undergo structural changes during the MCM2-7 hexamer/OCCM/DH/CMG transition. One review mentioned this (Riera et al 2017), but more data have been published since then.

Extended data 7: Please show one specific example of the horseshoe/helix conformation in isolation to clarify what structural change happens. This isn't easy to see in the figure, as several elements overlap.

Version 1:

Reviewer comments:

Reviewer #1

(Remarks to the Author)

The authors addressed all of my comments satisfactorily and I recommend that this manuscript is published in Nature Communications.

(Remarks on code availability)

The authors provide appropriately commented, well written bash scripts for dihedral and RMSD calculations with Pymol.

Reviewer #2

(Remarks to the Author)

I am satisfied that the authors have addressed my concerns. I recommend publication.

(Remarks on code availability)

Reviewer #3

(Remarks to the Author)

The authors have done an excellent job, and I am happy with the revised manuscript.

(Remarks on code availability)

We thank the reviewers for their careful consideration of our manuscript and suggestions for its improvement. Beyond their helpful suggestions, the reviewers raised profoundly important questions for us to consider, and our responses to these have led us to reframe the manuscript to emphasize an aromatic wedge mechanism for melting DNA that is conserved in archaea, eukaryotes, and DNA tumor viruses. We are grateful to the reviewers for pushing us to consider these questions. Key insights for this revision were obtained by calculation of RMSD of all 6 permutations of our structures (particularly Class 2b) against all eukaryotic MCM structures of the PDB. This analysis identified additional comparative structures and a consistent arrangement of the MCM ring when melting DNA that involves subunit-specific, conserved aromatic residues that we now describe as an “aromatic wedge”. Further details on the RMSD comparison are provided in specific responses below. The computer scripts and output for this analysis are provided as a supplemental archive file for interested reviewers to evaluate.

The reviewers had significant agreement on several points for improvement, including figure clarity, ATP site density figures, and the suggestion to incorporate human MCM among the structural comparisons. We have revised the manuscript based on these comments as detailed in our specific responses to each review below. Since the original submission, we have reversed the structural classes that we describe as “staggered” and “eclipsed.” This originally referred to the positions of vertically oriented helices of each tier, but we did not meaningfully discuss either helix. We now use “staggered” and “eclipsed” based on the positions of the underlying domains, which is more intuitively meaningful. We quantify the extent of such alignment with defined dihedral angles between the tiers and compare these to all eukaryotic MCM structures of the PDB. The computer scripts and output for this analysis are provided as a supplemental archive file for interested reviewers to evaluate.

We have also improved the structure of the 3 heptameric complexes by using a larger diameter focus mask during 3D classification. Although the heptameric structures are not a major focus of the manuscript, the modified 3D classification improved the DNA features of the map. The heptameric complexes have a larger diameter than the hexameric complexes, but our earlier treatment applied the same cylindrical focus mask that was used for the hexameric complexes. The EM maps for the heptameric complexes showed improved DNA density with the application of a larger diameter focus mask. The DNA appears to reside against one side of the channel rather than in its center, and this was more fully enclosed by the larger diameter focus mask. New PDB validation reports are uploaded. All other structures (each hexameric MCM and not heptameric) are unchanged from the original submission. The improved DNA density for the heptameric structure for the merged particle structure is shown below:

From original submission, Fig. S8:

Revised, now Fig. S15:

Responses to the comments of each reviewer are provided below.

REVIEWER COMMENTS

Reviewer #1 (Remarks to the Author):

Rasouli, Myasnikov and Enemark report a sheer number (18) structures of a duplex DNA bound protein chimera with a fusion of the Sulfolobus MCM N-terminal oligomerisation domain and the Pyrococcus MCM ATPase motor domain. Enemark has shown previously that this protein is catalytically active. Its use in structural biology studies already provided insights that proved precious for the field over time. The current work constitutes an important addition to this body of work. Here the authors report various mode of duplex DNA engagement for a single MCM ring. While the authors propose a mechanism whereby the structural transitions in this single hexamer could lead to DNA melting, they choose not to acknowledge some recent work on the mechanism of loading of a single MCM ring, and the structures of a fully loaded single hexamer around duplex DNA, from both human and yeast MCM (though, admittedly, the latter is only a bioRxiv uploaded manuscript). The recent identification of a single duplex DNA loaded MCM in humans was a big deal (with two high profile papers published simultaneously). The discovery that this is a stable intermediate also in archaea is remarkable and makes these data worth publishing in Nature Communications. That said, a rework of the introduction and discussion is recommended, which will provide a proper framework for the discoveries reported. Specific points are detailed below.

We thank the reviewer for encouraging comments and enthusiasm for the consistency between the archaeal structures and eukaryotic ones. The manuscript now cites recent loading structures and eukaryotic single-hexamer:DNA structures. The central question posed by the reviewer is which structures are most appropriate for comparison. To address this objectively, we now compute an RMSD of each permutation of our structure against each eukaryotic MCM ring in the PDB. Based on this analysis, we identified two

strong yeast CMG comparative structures (PDB 7PMN and 7PMK; now Fig. 6b and Fig. 6d), and we are now very confident to describe each of these structures as generating DNA melting. The RMSD analysis is detailed further in our response to point 1.

A logical extension of this question is why encircled double-stranded DNA appears intact in some complexes with MCM and melted in others. We show structurally that melting occurs based on subunit-specific, conserved aromatic residues of MCM that we term an “aromatic wedge.” This aromatic wedge and its role in DNA melting is conserved in our archaeal MCM structures, the comparative eukaryotic CMG structures, and in a recent structure of the more distantly related SV40 Large T-antigen (PDB 9KAE, shown in Fig. 7c), which also contains AAA+ ATPase domains and forms a double-hexamer to initiate unwinding DNA for replication. We have revised the focus of the manuscript and given it a new title in providing answers to the reviewer’s incisive questions. Further details are provided in response to point 3.

Major issues

1. Is it correct to define Class 2 as a DNA melting intermediate, considering the discovery from Steve Bell’s lab (Champasa et al 2019) that a yeast MCM variant that cannot homodimerise can form CMG but not open DNA? Would it not be best to compare this structure to the human MCM single hexamer (Weissmann et al 2024 and Yang et al 2024)?

We are grateful for these two questions. First, we are grateful for the opportunity to add the Champasa citation and improve the clarity of our manuscript. As noted by the reviewer, the Champasa et al 2019 article describes a yeast MCM variant that does not robustly form a double-hexamer of two Mcm2-7 rings but does form CMG. On the specific point on whether this variant can “open DNA”, the article shows that the variant can generate initial DNA melting, but it does not produce extensive origin unwinding. We feel that the confined DNA melting that we report for both Class 2 structures is consistent with the initial melting described in Champasa et al 2019. We have added this reference and spelled out that we feel our structures correspond to initial melting and not extensive origin unwinding. We also feel that it is correct in literal definition to describe the lost base-pair interactions of the Class 2 structures as “melting”. As described further below, disorder of nucleotides following loss of base-pair interactions is a common theme in DNA-melting structures and is observed in multiple newly added comparative structures (PDB 7PMN of Fig. 6b, PDB 7PMK of Fig. 6d, and PDB 9KAE of Fig. 7c).

Regarding the second question on structural comparison choice, the density of PDB 7Z13 is very clear among the features that were the focus of our initial submission, and we were intrigued by its strong visual similarity with Class 2b. The reviewer’s question of whether a different structure might provide a more suitable comparison for the Class 2 structures led us to objectively verify that we were not comparing to PDB 7Z13 solely out of admiration. For this verification, we downloaded a report from the PDB with all structures meeting the search term “mcm” and a 200 kDa molecular weight cutoff. For each, we generated a

hexameric MCM structure consisting of six 152-amino acid ATPase core structures (see Meagher et al IJMS 2022. doi: 10.3390/ijms232314678). The RMSD of this structure with each of the six permutations of the corresponding 6 X 152-amino acid hexameric ATPase core of Class 2b was calculated and sorted to compare the collective disposition of the six MCM ATPase domains as a unit (see Supplementary Fig. 10 for illustration of the process including the six settings of Class 2b for candidate alignments). Altogether, 77 structures were compared in 6 settings for a total of 462 comparisons. The top 30 RMSD alignments (Supplementary Table 1) were visually inspected. The alignment of PDB 7Z13 with Class 2b in the setting used for comparison in the initial submission (with chain E corresponding to Mcm3) was among the top 30, and we would like to retain this comparison in Fig. 6e.

This analysis also revealed two structures with striking similarity to the Class 2 structures of our manuscript that we now include among the comparisons of Fig. 6. PDB 7PMN (Fig. 6b) is similar to Class 2a with two tight interfaces (ANP-bound) and two unpaired DNA bases. PDB 7PMK (Fig. 6d) is similar to Class 2b with three tight interfaces (ANP-bound) and four unpaired DNA bases (structures from doi: 10.1038/s41586-021-04145-3). These structures align in the same setting as PDB 7Z13 with Class 2 chain E matched to Mcm3. Importantly, the structures of PDB 7PMN and PDB 7PMK were determined with a substrate that does not have an exposed 5'-end, and hence the DNA in these structures represents *bona-fide* melted DNA. As with the Class 2 structures that we describe, DNA melting occurred at an end of the DNA molecule, and nucleotides not bound by MCM that have lost base-pair interactions are disordered.

Notably, 28 of the top 30 RMSD alignments are CMG structures. One loading complex that does not contain Cdc45 or GINS (PDB 9GJW) was among the top 30 and is now compared to Class 2b in Fig. 8b. This structure/alignment was similar to a structure of CMG bound to a G-quadruplex/ssDNA hybrid substrate, and this is now included in Fig. 8a.

The specific subunit arrangements (Table 1) of our presented structures and the comparative structures now form the basis for a sequential melting mechanism illustrated as a cartoon in Fig. 9a to correspond to the initial melting described in the Champasa et al 2019 article referred to above.

2. The authors observe Class 2 where the leading but not the lagging strand template is visible. They interpret this as a melting intermediate. This could also reflect the fact that the leading strand is visible because it is stabilised by pore loop contacts. This is indeed the case for a helicase loading intermediate (single loaded yeast MCM hexamer) described in a bioRxiv manuscript by Miller, Diffley and colleagues. The PDB coordinates (8RIG) can be downloaded from the PDB already. This is an intermediate on the path to double hexamer formation. In the final product (the DNA is not melted).

We believe that interactions between the pore loops and this strand directly assist its melting from the complement. PDB 8RIG is among the structures used in the above RMSD analysis. The lowest RMSD for PDB 8RIG with Class 2b was ranked 46 (in the setting where

Class 2b chain E is matched to Mcm6). As this was not among the top-30, we do not consider it as strong a match for comparison as the ones that we have selected.

3. Along the same lines, I feel the authors should do a better job at drawing a parallel between the mechanism of MCM loading onto duplex DNA and the mechanism of DNA melting by the CMG. In this respect, I note that no reference was made to recent work published in Nature Communications by Butryn et al (2025), which describes a single-loaded yeast hexamer in different duplex DNA binding states, captured by using inactivating ATPase mutants. These structures show a DNA engagement state similar to that of the DNA-melting (or single-stranded DNA translocating) CMG, concluding that the mechanism of MCM interaction and translocation along DNA is always the same, and implying that melting occurs when two CMGs are facing one another. I feel this is another example of a missed opportunity to show how the structure of an archaeal single loaded hexamer on duplex DNA is relevant to mechanistic studies in eukaryotes (both for DNA melting, as stated in this manuscript, but also for helicase loading).

The central question that the reviewer raises for consideration is why MCM-encircled double-stranded DNA is intact for some complexes (such as the noted loading complexes) and melted in others (such as PDB 7Z13 or Class2b). We now discuss that we feel these differences arise from the **aromatic wedge** (see below) and its precise placement on DNA (its specific “setting”). We specifically favor the setting illustrated when Class 2b chain E aligns to Mcm3 (as in PDB 7Z13, and 7PMK) as a melting conformation. We note that this structural “melting conformation” appears to be especially stable because 21 of the top-30 alignments are in “setting 1”, which aligns Class 2b chain E with Mcm3. Another illustration of the stability of this form is its abundance as “staggered Form II” in the new Fig. 4.

This setting develops tight ATPase interfaces among the Mcm5, 2 and 6 subunits, which are precisely the subunits that stack an aromatic residue on the bases and sugars of the melted DNA. Further, these subunits specifically conserve the aromatic residue (“aromatic wedge”) while the other subunits are nearly always **not** aromatic (Fig. 7d; also see Supplementary Figure 3 of Meagher et al 2019). As a result, only eukaryotic Mcm2-7:DNA:ATP arrangements of Setting 1 (that match the setting of PDB 7Z13) appropriately position the aromatic residues of the 3 adjacent subunits to assist in DNA melting. Other settings do not optimally position each aromatic group for melting and therefore are not expected to melt DNA as robustly. Due to the homo-hexameric construction of the archaeal MCM hexamer, all settings would position the conserved aromatic group equivalently.

We previously showed that the corresponding aromatic residue in a homo-hexameric archaeal MCM (SsoMCM Y386, Meagher et al 2019, doi: 10.1038/s41467-019-11074-3) stacks on the bases and sugars of single-stranded DNA and showed that an aromatic residue is not needed in this position to unwind DNA in a strand displacement assay based on mutation to alanine (Supplementary Figure 4 of Meagher et al 2019). Notably, this

mutation eliminates the aromatic group from all 6 subunits of the homo-hexamer. We further noted that the position of the aromatic group is similar to that of a conserved histidine of the related hexameric helicase E1 from papillomavirus (Compare h2i Y386 to ps1b H507 in Supplementary Figure 2 of Meagher et al 2019, doi: 10.1038/s41467-019-11074-3). This was especially intriguing to us because Arne Stenlund had previously mutated E1 H507 to 8 possible residues (A, V, L, R, M, N, F, and Y) did not prevent unwinding DNA in a strand displacement assay, but several melting and replication activities critically relied on an aromatic residue in this position (doi: 10.1016/j.molcel.2007.02.009). Of the mutants tested, only H507F supported in vivo replication in transient DNA replication assay, and only the aromatic mutants (H507F and H507Y) supported in vitro replication in a cell-free replication system. Based on permanganate reactivity, only H507F melted DNA in the T-rich region that is anticipated to be located at the E1 ATPase tier, and only H507F provided unwinding in an origin unwinding assay (doi: 10.1016/j.molcel.2007.02.009). Accordingly, we postulated (Meagher et al 2019) that the MCM h2i aromatic residue and the E1 ps1b histidine have similar roles in melting DNA but are dispensable for strand displacement during processive unwinding.

We now strengthen this analogy based on a recent structure of SV40 T-antigen, a close relative of E1, bound to melted DNA (PDB 9KAE, Fig. 7c). In this structure, the aromatic histidine side-chain stacks on the bases of the melted DNA similar to the aromatic residue of archaeal MCM (our Class 2 structures) and eukaryotic Mcm5, 2, and 6 (PDB 7PMN, PDB 7PMK, and PDB 7Z13). Further, the nucleotides not bound by T-antigen that have lost base-pairing interactions are disordered, analogous to the behavior of the melted DNA of our Class 2 structures and the eukaryotic CMG structures of PDB 7PMN and 7PMK.

Based on this collective comparative analysis, we have adjusted the focus of the article and the article's title to explicitly describe a role for an aromatic wedge in DNA melting in archaea (Class 2 structures of this manuscript), eukaryotes (PDB 7PMN, 7PMK, 7Z13), DNA tumor viruses (E1 and SV40 T-antigen, PDB 9KAE).

Smaller issues

1. The authors claim that no contacts exist between MCM and DNA in the class 1 structure. They should clarify how the DNA register is established such that duplex DNA can be averaged in the structures. If not contacted by the ATPase domain, duplex DNA must surely be engaged by elements of the N-terminal oligomerisation domain. Otherwise, DNA would be lost from the central channel or only result in a featureless sausage through single-particle averaging. You could refer to studies on the FANCI-FANCD2 DNA encircling ring by Alcon et al Nature 2024, Fig. 4a and Fig. 4b for comparison.

The DNA does not appear bound in the Class 1 structures with specific interactions. When inspecting the maps, the most prominent approach of the protein to DNA appears at one of the helix-2-insert hairpins. However, this appears too distant for a direct interaction such as a lysine-phosphate interaction. It strikes us more like a dipole-dipole interaction that

might contribute to a weakly preferred orientation. During refinement and classification of the structures, the DNA did appear as a featureless sausage. We now show this in Supplementary Fig. 6 as part of the cryo-EM workflow and illustrate below:

The DNA was subsequently subjected to 3D classification with a focus mask that encompasses the central channel. For the Class 1a structure that is essentially C6-symmetric at the protein, this classification would be expected to identify six different permutations of DNA structure. These structures would each span a 60° rotation range about the DNA helical axis if the DNA is a regular helix and its orientation is fully random. For illustrative purposes, hypothetical versions of such DNA maps were constructed by rotating a B-DNA molecule through a 360° range about its helical axis in 1° increments, and composites of the structures were used to generate maps in chimera at 6 Å resolution (for example: molmap #0-359 6). The resulting maps are depicted below and equally weighted by setting the threshold level according to the number of contributing models. Rotation of a full 360° about the B-DNA helical axis generates the featureless sausage described by the reviewer. Individual 60° classes resemble the final model, indicating that the DNA of Class 1a is likely close to undistorted B-form and that this DNA probably does not strongly interact with the MCM ring because such interaction would likely provide a more featured DNA appearance than the modest resolution of the 60° wedge examples.

In the case of the Class 1b and 1c structures where the protein complex is asymmetric due to a disordered ATPase domain, the initial map has minor double-helix features (see above from Supplementary Fig. 6), indicating a slight orientational preference. As with Class 1a, the most prominent approach between the protein and DNA appears at one of the helix-2-insert hairpins—specifically the hairpin of a subunit adjacent to the disordered ATPase domain. However, this hairpin also appears too distant from the DNA for a direct interaction (such as a lysine-phosphate interaction). As with Class 1a, this strikes us more like a dipole-dipole interaction that might contribute to a weakly preferred orientation.

For these structure classes, we selected two classes from 3 using the cylindrical focus map (see Supplementary Fig. 6). For this case, a random DNA orientation can be emulated by rotating a DNA molecule through larger ranges as illustrated below where two 101-degree ranges provide hypothetical maps with discernable major and minor grooves and approximately align one DNA strand in opposite polarity as in the structures of Class 1b and Class 1c. The remaining hypothetical class has a double-helix appearance without assignable major and minor grooves. Thus, the overall appearance of our structures suggest that the DNA orientations have orientation preferences that are likely very small but slightly better than random.

2. Heptameric MCMs had been previously observed in studies by Jose' Maria Carazo, Silvia Onesti and Laura Spagnolo. Their work should be cited.

Yes- this should be cited. Thank you for pointing this out. References #47, 74, 81, and 82 of the revised submission now refer to heptameric structures. One suggested article (Cannone et al, 2017; DOI: 10.1038/srep42019) describes an octamer rather than a heptamer.

3. The resolution of the structures reported is high enough for discriminating nucleotides in the ATPase pockets but the authors showed not to show them. I think this should be rectified.

We have prepared a figure to illustrate the triphosphate regions of the maps (Fig. 2) that also demonstrates one of the intersubunit distances that we use to assign a “tight interface”. In the presented maps, the triphosphate is generally clearly visible for the tight interfaces while triphosphate density is less clear for the loose interfaces. We have modeled weaker cases as ADP molecules, but we note that we are not able to differentiate disordered γ -phosphate from hydrolyzed γ -phosphate, and so we emphasize that the molecule assigned in the PDB model does not indicate that hydrolysis must have occurred.

4. Line 37 page 2 “Similar, but different subunits” sounds odd. To me, similar implies that they are different, otherwise they would be identical.

We have deleted “but different”

5. Figure 1 is very similar to Extended Data Figure 2 and 3. Figure 2 appears very similar to Extended Data Figure 4 and 5. Could the authors check whether any panels have been mistakenly misplaced?

We have inspected all placed panels of:

Fig. 1/Supplementary Fig. 8/Supplementary Fig.9
Fig. 5/Supplementary Fig. 11/Supplementary Fig. 12
Fig. 6/Supplementary Fig. 11/Supplementary Fig. 12

The correct panels are placed (figure numbers above are the new figure numbers). Each are viewed from the same perspective, and the structures are highly similar. In particular, the merged particles structures are derived from the particles that generated the individual DNA species structures— and so similarity is expected. One area where differences can be perceived is in the cutaway views that illustrate encircled DNA (Rightmost panels of Fig.1/Supplementary Fig. 8/Supplementary Fig.9). The DNA1 structures have DNA order that persists further below the ring than those of DNA2. The corresponding merged particle structures show an intermediate length of order. On the other hand, the DNA2 structures are higher resolution than the corresponding structures of DNA1. The merged particle structures appeared to provide a good compromise between structure resolution and extent of DNA visualization.

6. Page 3 line 72 “GINS recruitment provides 1.5–2 turns of DNA unwinding at each CMG” is incorrect. Douglas 2018, which the authors cite in this passage reads “each MCM hexamer untwists around 0.6–0.7 turns of DNA”.

Thank you for pointing this out. We misinterpreted a different sentence from the Douglas et al manuscript as the unwinding number (“Thus, each of the two activated CMG helicases constrains approximately 1.5-2 helical turns of unwound DNA.”). We have changed the cited number.

Reviewer #2 (Remarks to the Author):

A large number of biochemical and cryo-EM studies in the past few years have focussed on the molecular characterisation of the initiation stages of DNA replication. Progress in this area has been significantly advanced by the ability to reconstitute the process in vitro using purified yeast proteins, and solve structures of various intermediate states. Although much is now known about this process, a complete molecular explanation of the very first stages of DNA melting is still lacking. This manuscript focusses on this key question, using the archaeal homo-hexameric MCM replicative helicase as a model system.

The authors present multiple cryo-EM structures of an archaeal MCM single hexamer bound around DNA. Their sample preparation method facilitates the capture of multiple different forms of the MCM-DNA complex, either containing B-form double stranded DNA, or in various states that the authors propose mimic the first stages of replication origin melting. By integrating these structures with previously published structures of archaeal and yeast MCM complexes, the authors propose a compelling model for DNA melting, involving sequential conversion of inter-subunit interfaces from loose to tight states. The manuscript is well structured and written, and clearly explains some quite complex structural conclusions and hypotheses. Further interrogation of the structural data using functional experiments (e.g. with specific mutants in vitro or in vivo) is completely lacking. Overall, however, given the high quality of the structural data and the general interest in this area of the replication field, I would suggest that the structures alone are sufficiently interesting to warrant publication without functional validation, particularly given the difficulty of such experiments in the archaeal system.

We thank the reviewer for supportive comments that the model is compelling and that the manuscript has achieved clarity in describing structural concepts that are potentially complex. We strived to make the writing accessible to both experts and non-experts, and so these comments are especially gratifying.

Major points:

- The author’s models are heavily dependent on nucleotide-dependent transitions between loose and tight states at inter-subunit interfaces in the MCM ring. To this reviewer, it was unclear whether the authors actually had high enough resolution in their structures to model nucleotide in any of the MCM active sites? If not, this should at least be stated.

We have prepared a figure to illustrate the triphosphate regions of the maps (Fig. 2) that also demonstrates one of the intersubunit distances that we use to assign a “tight interface”. In the presented maps, the triphosphate is generally clearly visible for the tight

interfaces while triphosphate density is less clear for the loose interfaces. We have modeled weaker cases as ADP molecules, but we note that we are not able to differentiate disordered γ -phosphate from hydrolyzed γ -phosphate, and so we emphasize that the molecule assigned in the PDB model does not indicate that hydrolysis must have occurred.

- The model in Fig. 5 is dependent on sequential transitions in inter-subunit interfaces around the MCM ring. One tight interface appears to favour the conversion of the neighbouring interface from loose to tight. Can the authors explain how this would work? Is there a structural explanation for why ATP binding at one interface would, for example, favour ATP binding at the adjacent interface?

The directional order illustrated in Fig. 5 (now numbered Fig. 9) was originally based on the presumed directional order that the complex uses in a rotary DNA translocation mechanism (Meagher et al 2019, doi: 10.1038/s41467-019-11074-3). The DNA-binding and ATPase site differences between Class 2a and Class 2b are readily distinguished, and a structural transition from Class 2a to Class 2b would involve tightening one subunit interface that is adjacent to two consecutive tight interfaces (See Fig. 6 or Fig. 9). The tightened interface could occur at the first empty interface clockwise or counterclockwise. Considered at the ATP site level, the ATP site that is tightened could involve the Walker-A site of one of the subunits participating in tight binding or the arginine finger of one of these subunits. These scenarios are not distinguishable for archaeal homo-hexameric structures because the subunits are identical in sequence. In the absence of contrary information, we presumed that the directional order for tightening ATPase sites would be consistent with that used for rotary DNA translocation and thus occur counterclockwise in the view of Fig. 9a. The newly added comparative structures of PDB 7PMN and 7PMK definitively resolve the directional order because they derive from heterohexameric structures with distinguishable subunits. These structures illustrate that the tightened interface is at the counterclockwise side (3/5 interface tightened rather than 6/2), and hence the directional order of ATP-binding during melting is indeed identical to the directional order used for DNA translocation.

- Recent structures have demonstrated that a small section of DNA, positioned at the inter-hexamer interface, is already distorted in the human MCM double hexamer (e.g. PDB:7W1Y). Thus, unlike in yeast, melting (or at least distortion of the duplex DNA) has already been initiated before CMG assembly. On the one hand, the authors make almost no mention of the human structures. The authors repeatedly refer to ‘eukaryotic’ structures, when in reality they are only referring to yeast. Given that a major selling point of this manuscript is the derivation of a generalised model of MCM-driven melting, the authors need to properly consider the implications of the human MCM structures for their models. For example, how do the authors envisage that the initial bubble of distorted DNA trapped near the N-tier of MCM (in the human structures) is converted to melted DNA in the MCM C-tier of each single hexamer, as has been demonstrated previously by the Diffey-Costa groups, as well as in this manuscript?

The reviewer raises a good point to enhance the generality of eukaryotic comparison, and we have added PDB 7W1Y to what is now Fig. 5.

The 7W1Y to 7Z13 transition has been very intriguing to ponder, and we are grateful to the reviewer for providing the motivation to do this. PDB 7W1Y is a highly informative structure that beautifully illustrates the structure of a human double-hexamer encircling DNA in a conformation poised to proceed to replication initiation and provides exquisite resolution for the constituent components. We do not intend our response below to the reviewer's question about the N-tier bubble to diminish the remarkable significance or technical achievement of the PDB 7W1Y structure.

The bubble at the N-tier of PDB 7W1Y seems to at least partly derive from the double-hexamer architecture because the corresponding single-hexamer structures (PDB 8S0A and PDB 8W0E) are quite similar in underlying hexamer:DNA arrangement, but they do not exhibit melted DNA at the N-tier. The N-tier bubble in PDB 7W1Y appears to derive from simultaneous interaction of each constituent hexamer C-tier with DNA in conjunction with a double-hexamer interaction that enforces a greater distance between the C-tiers than undistorted B-DNA allows. If any of these three interactions are lost, the additional degrees of freedom potentially will allow the DNA to re-anneal. For multiple arguments below, we feel that at least one of the hexamers of PDB 7W1Y must release its grip on DNA (while continuing to encircle DNA) and that such release would generally permit the N-tier DNA bubble to re-anneal.

First, one of the initial steps towards replication initiation is exhaustion of nucleotides from the MCM ATPase sites. Based on several MCM:DNA structures and arguments presented in this manuscript, we fundamentally associate DNA-interaction with the presence of nucleotides at the corresponding ATPase sites to create a "tight" interface to appropriately position DNA-binding modules of adjacent subunits for binding DNA (now illustrated in Fig. 3 that demonstrates helix versus horseshoe). Exhaustion of all nucleotides from the MCM ATPase sites is expected to convert all interfaces to "loose" with concurrent loss of interactions with DNA (analogous to the Class 1a structure that we present, see Fig. 5). Hence, exhaustion of nucleotides is predicted to release the grip of both hexamers on DNA. Although the hexamers would continue to encircle DNA, the lost grip would likely allow the N-tier bubble to re-anneal.

Second, a fundamental global difference between the 7W1Y and 7Z13 structures strongly suggests that at least one hexamer dissociates its grip on DNA in order to transition between the two arrangements. Each structure is effectively 2-fold symmetric with a hexamer:hexamer dyad co-aligned with a DNA dyad, but in ***opposite directions***. The 7W1Y dyads place the DNA major groove at the Mcm3:Mcm3' side while the 7Z13 dyad has the DNA minor groove at the Mcm3:Mcm3' side (see below). Interconversion of these forms very likely involves at least one hexamer losing its grip on DNA. The forms can interconvert by either: 1) rotating each hexamer 180° about the DNA helical axis (both

rotating the same direction or each rotating in opposite directions); 2) by sliding the hexamers one-half helical turn along the helical axis; or 3) by a combination of rotation and slide of the hexamers with respect to the DNA helical axis. In each scenario, at least one hexamer would need to disengage its binding grip on DNA (detailed below), which in turn would likely allow the N-tier bubble to re-anneal.

A more detailed explanation that at least one MCM hexamer would need to release its grip on DNA can be viewed with the DNA-binding register of the two complexes (see illustration below). For this comparison, the author-provided nucleotide numbers are provided along with the number of nucleotides from the DNA dyad, which provides a consistent reference for comparison. As illustrated, nucleotides at dyad+16 and dyad+17 are bound at Mcm7 in PDB 7W1Y, but dyad+16 and dyad+17 are bound at Mcm3 and Mcm5 and PDB 7Z13. Two scenarios are possible with this difference: 1) the DNA disengages binding to Mcm7 and re-engages binding at Mcm5 and 3; or 2) the DNA dyad that aligns to the double-hexamer dyad changes during the transformation (e.g. the midpoint of the two MCM hexamers translates with respect to DNA). The protein disengagement of Scenario 1 implicitly provides a period where hexamers lose their grip on DNA to likely allow a capacity for the N-tier bubble to re-anneal.

For the second scenario, we feel it could be possible for the MCM double-hexamer to translate along the DNA axis based on the translocation activity of **one hexamer**. In this action, an active translocating hexamer could maintain its grip on DNA, but the second hexamer would need to release its grip on DNA and passively slide (perhaps viewed as being pushed by the first hexamer). This scenario therefore also involves at least one hexamer losing its grip on DNA and thus provides the capacity for the N-tier bubble to re-anneal.

With a likely loss of at least one hexamer:DNA contact, it strikes us as too demanding for the replication initiation mechanism to require that the N-tier bubble must not re-anneal in

order for replication initiation to be successful. Once the two hexamers have separated, we feel that it would not be possible to regenerate this N-tier bubble if these nucleotides were to re-anneal unless the overall double-hexamer is regenerated. In contrast, if melting is based on an aromatic wedge within single hexamers (as in our structures and in PDB 7PMN and PDB 7PMK), melting and re-annealing can potentially occur repeatedly prior to successful advancement to extended origin unwinding and replication initiation.

- In the yeast system, the ejection of ADP from MCM and the initiation of origin melting is dependent on the conversion of MCM double hexamers into pairs of CMG helicases, driven by multiple so-called firing factors (Douglas et al., Nature, 2018).

Given this, it is quite surprising that the authors can observe DNA melting by simply loading an MCM single hexamer onto double-stranded DNA in ATP γ S. Do the authors think this reflects a biologically meaningful difference between the ways in which melting is initiated in archaea and eukaryotes?

We believe that melting at the MCM ATPase domain occurs fundamentally the same way in archaea and eukaryotes and is derived from an aromatic wedge that is now a central focus of the manuscript. All eukaryotic structures of MCM with DNA melted at the MCM ATPase domain are CMG structures. The aspect that the reviewer likely finds surprising is that the Class 2 structures reported in our manuscript do not have Cdc45 or GINS but nevertheless exhibit melting. Our newly added RMSD-based structure comparisons indicate that the Class 2 structures with melted DNA most closely resemble eukaryotic CMG structures. Of the top-30 scoring RMSD eukaryotic comparative structures, 28 are CMG structures. Thus, it appears to us that although the Class 2 structures do not possess Cdc45 or GINS, they somehow structurally mimic a CMG arrangement for the collective ATPase tier.

- The authors prepare their samples in the slowly hydrolysable ATP analogue ATP γ S. Presumably the origin melting they observe should be dependent on ATP hydrolysis? What do the authors observe if they prepare their complexes in non-hydrolysable nucleotide (e.g. AMP-PNP)?

We ascribe the different conformations observed to whether/how ATP γ S is bound at the respective ATP sites. We believe that the limited melting in our Class 2 structures (and in PDB 7PMN and PDB 7PMK, which are both with AMP-PNP) is based on ATP-binding and that hydrolysis is not necessary. This is not processive DNA unwinding that would require ATP hydrolysis in a rotary cycle.

- Although the authors are primarily using the archaeal system as a model to understand general principles of MCM function, the lack of discussion regarding the biological mechanisms of DNA replication initiation in archaea seemed an oversight. The authors should describe what is known about the various steps in MCM loading and activation (e.g. as part of the introduction). This is also relevant to point 4 (above).

We have added additional details to the introduction about replication initiation in archaea.

Minor points:

- Lines 312-315: the authors note that the archaeal structure with 4 tight interfaces also a disordered subunit, potentially providing an exit route for strand exclusion. Is the same true for eukaryotic CMGE (PDB: 7QHS)? If not, does this reflect mechanistic differences between the archaeal and eukaryotic systems?

In order to focus on an aromatic wedge melting mechanism, we have removed the description of an archaeal MCM ring encircling ssDNA where one ATPase domain is disordered. However, to answer the reviewer's question, we believe that the mechanism that the lagging strand exits the ring is not well-understood in any system, and we think this represents a highly intriguing question. Without contrary information, we presume that archaea and eukaryotes use similar mechanisms for such strand exit. The open interface of the 4-tight interface structural complex is an intriguing potential candidate for part of how a strand might exit. An analogous eukaryotic structure with a mobile ATPase domain has not been described to our knowledge, but may exist. We now only briefly refer to the 4-tight interface structure in a short lagging strand exit section.

- Why do the authors use a chimeric MCM fusion, with different domains from different archaeal species?

We first generated this chimeric fusion to investigate its ability to unwind DNA in a strand displacement assay. We empirically found it to be extremely robust as a hexamer and active for DNA-unwinding in a strand displacement assay. The construct also proved very useful for structural studies because we were able to determine a crystal structure of the MCM hexamer (Miller et al 2014, doi: 10.7554/eLife.03433). This structure was cited as the template model in the first high-resolution cryo-EM structure of eukaryotic Mcm2-7 (Li et al 2015, doi: 10.1038/nature14685). In over 10 years of work with several archaeal MCM constructs for structural studies by X-ray crystallography, we have only successfully crystallized two constructs in a hexameric form that contains both the N-terminal domain and the AAA+ domain: this *Sso-Pf*MCM chimera and a modified *Sso*MCM construct (Meagher et al 2019, doi: 10.1038/s41467-019-11074-3). We have since used cryo-EM to identify and analyze additional MCM:DNA forms for both constructs, and these allow direct comparison with the original crystal structures. We previously reported cryo-EM structures of the *Sso*MCM construct encircling single-stranded DNA, which allowed us to identify additional states of DNA translocation (Meagher et al IJMS 2022. doi: 10.3390/ijms232314678). In the current manuscript, we were motivated to elucidate how the archaeal MCM hexamer interacts with encircled double-stranded DNA, something we have never been able to capture by X-ray crystallography.

- More detail should be given on the conditions (concentrations, buffers, times, temperatures) used for preparing the samples before grid preparation

We have added experimental details for protein purification and for generation of the sample of cryo-EM analysis. A comprehensive cryo-EM workflow is now provided as Supplementary Fig. 1-7.

- The different coloured circles that the authors use to explain the different subunit conformations was barely visible on my printed version

We have removed the colored circles from the main figures and have added dashed outlines to highlight structural features.

Reviewer #3 (Remarks to the Author):

Rasouli et al have investigated the DNA unwinding ability of a hybrid archaeal MCM helicase. The authors employed double-stranded DNA for their cryo-EM analysis of the protein-DNA complexes. ATPγS was employed to slow down ATP-hydrolysis. The sample yielded multiple complex variants. Two classes were observed – one, where dsDNA is observed in the NTD and CTD, the MCM CTD and NTD are twisted, and ATO occupancy is low. The second class, with partially unwound DNA in the CTD, the MCM CTD and NTD are not twisted and nucleotide is observed in several subunits. Within each class, several variants are observed. Class 1 variants detail CTD flexibility, while class 2 variants have different numbers of nucleotides bound and unwound nucleotides. The work is exciting, as it shows the stepwise activation of an MCM hexamer. One could complain that the MCM complex is a hybrid between two different species, but I would argue that the construct is well suited to decipher a mechanism. Nevertheless, some justification should be given. The structure is well described, but figures must be improved to highlight key structural changes, especially for a general audience. Also, the processing workflow is missing and the identified changes could benefit from some verification. The comparison to eukaryotic MCM is well done.

We thank the reviewer for supportive comments. In particular, the reviewer's comment that the comparison to eukaryotic MCM is well-done is gratifying because we strived to create a cogent comparison to eukaryotic MCM. We feel the revised manuscript has strengthened this comparison.

We first generated this chimeric fusion to investigate its ability to unwind DNA in a strand displacement assay. We empirically found it to be extremely robust as a hexamer and active for DNA-unwinding in a strand displacement assay. The construct also proved very useful for structural studies because we were able to determine a crystal structure of the MCM hexamer (Miller et al 2014, doi: 10.7554/eLife.03433). In over 10 years of work with several archaeal MCM constructs for structural studies by X-ray crystallography, we have only successfully crystallized two constructs in a hexameric form that contains both the N-

terminal domain and the AAA+ domain: this *Sso-Pf*MCM chimera and a modified SsoMCM construct (Meagher et al 2019, doi: 10.1038/s41467-019-11074-3). We have since used cryo-EM to identify and analyze additional MCM:DNA forms for both constructs, and these allow direct comparison with the original crystal structures. We previously reported cryo-EM structures of the SsoMCM construct encircling single-stranded DNA, which allowed us to identify additional states of DNA translocation (Meagher et al IJMS 2022. doi: 10.3390/ijms232314678). In the current manuscript, we were motivated to elucidate how the archaeal MCM hexamer interacts with encircled double-stranded DNA, something we have never been able to capture by X-ray crystallography.

Major comments:

Could the authors test their DNA unwinding model biochemically e.g. by mutating S357 or similar.

The reviewer raises an interesting point about S357 because this residue appears to play a prominent role in two aspects of the sequential melting that our manuscript describes—both in the development of tight interfaces and in DNA-binding among the subunits that adopt a tight interface. We have previously tested unwinding activities for mutants of the related archaeal SsoMCM (see Supplementary Figure 4 and Table 2 of Meagher et al 2019, doi: 10.1038/s41467-019-11074-3) as part of a structure-function report on DNA translocation and unwinding. The structure from the earlier report is now illustrated in Fig. 7a for comparison. Although the present manuscript is on the topic of melting rather than processive DNA unwinding, SsoMCM T369A mutants of the helix-2-insert hydroxyl group (T369 corresponds to S357 of the present structure, see Fig. 7a-b) are consistent with a role for this residue to function with neighboring subunits. In our 2019 manuscript, mutational analysis was performed in two contexts- 1) wild-type SsoMCM; and 2) a construct that lacks the C-terminal HTH domain (that is dispensable for unwinding) and incorporates a shortened linker between the N-terminal domain and the C-terminal domain (SsoMCM-GGSGGS- Δ C). In the absence of mutations, the modified construct showed much stronger cooperativity for unwinding than wild-type ($h=6.610 \pm 2.030$ versus $h=1.160 \pm 0.2619$). Among mutants tested, the T369A mutant was unique in that it could not be expressed and purified in the full-length protein context. The T369A mutant could be isolated in the more cooperative SsoMCM-GGSGGS- Δ C context, but with a vastly reduced cooperativity ($h=0.7792 \pm 0.2188$). Collectively, the results indicate a role for T369 in the cooperative function of MCM ATPase subunits and that added cooperativity of the SsoMCM-GGSGGS- Δ C context may partly compensate for the lost cooperativity upon mutation of residue 369 to alanine. We have added a description of the reduced cooperativity of SsoMCM-GGSGGS- Δ C T369A to the current manuscript.

The authors need to provide a detailed EM data processing workflow.

We have added a detailed workflow with intermediate images as Supplementary Fig. 1-7.

The description of the main structural changes e.g. eclipsed vs staggered need to be illustrated by a cartoon. The vertically oriented α -helices are mentioned but not clearly shown in any figure. The structural changes need to be conceptualised and shown in the form of a cartoon.

We attempted to mark the vertically oriented helices with partially transparent circles, and these clearly did not work well. Reviewer 2 also noted that these are barely visible. We now provide a staggered/eclipsed cartoon in Fig. 4. We have changed the fundamental description of eclipsed versus staggered to not focus on the vertical helices. The vertical helices were prominently different to us during structure determination, and we used them to distinguish the forms. The revised description focuses on overall domain positions, which are more mechanistically meaningful. The domain positions are also amenable to the dihedral calculations of Fig. 4. We feel that the quantified dihedrals are more meaningful and interpretable and allow comparison with heterohexameric eukaryotic Mcm2-7 complexes.

Minor comments:

In the context of the staggered and eclipsed conformation, the eukaryotic should be discussed more, here, MCM2-7 NTD/CTD undergo structural changes during the MCM2-7 hexamer/OCCM/DH/CMG transition. One review mentioned this (Riera et al 2017), but more data have been published since then.

This reviewer comment greatly strengthened the manuscript because it prompted the dihedral analysis of Fig. 4 to explore the scope of inter-tier conformations among eukaryotic Mcm2-7 structures. The reviewer is correct about the suggested analogy—thank you for pointing this out. We examined the inter-tier rotation of all 77 eukaryotic Mcm2-7 structures of the PDB used for the above RMSD calculations of the AAA+ hexamer. The inter-tier rotation was quantified by dihedral angles defined by consistent positions of the N-tier OB fold, the C-tier AAA+ fold, and two positions of the ring central channel. Five of the eukaryotic models were then excluded (8RWV, 6SKO, 8q6P, 8s0e, 8s0f) because they do not have a full set of 6 OB-folds and 6 AAA+ domains in the model. The 72 models remaining have two obviously distinct categories that we now classify as eclipsed (mean dihedral: -6 to -15) and staggered (mean dihedral: -29 to -38). The review article mentioned by the reviewer (Riera et al 2017) ascribes a major role for Cdt1 in generating a different conformation for OCCM than in DH or CMG. The dihedral analysis is fully consistent with such a role because 100% of the structures with Cdt1 are classified as eclipsed based on the dihedrals. We have added a plot of the complete dihedral analysis for each subunit of the 72 analyzed structures in Fig. 4. A movie illustrating the general theme of eclipsed and staggered among the presented structures and some eukaryotic structures is presented as Supplementary Movie 1.

Extended data 7: Please show one specific example of the horseshoe/helix conformation in isolation to clarify what structural change happens. This isn't easy to see in the figure, as several elements overlap.

We now illustrate the helix-horseshoe conformations for three of our structures in Fig. 3, highlighting the specific regions with dashed outline. These figures also illustrate an intersubunit distance that correlates the helix conformation. Along with the ATPase site distance of Fig 2, these two distances define whether we assign subunit interfaces as “tight” (Table 1).